# Recapitulation of selective nuclear import and export with a perfectly repeated 12mer GLFG peptide

Sheung Chun Ng[1], Thomas Güttler [1] & Dirk Görlich [1✉]

The permeability barrier of nuclear pore complexes (NPCs) controls nucleocytoplasmic transport. It retains inert macromolecules while allowing facilitated passage of importins and exportins, which in turn shuttle cargo into or out of cell nuclei. The barrier can be described as a condensed phase assembled from cohesive FG repeat domains. NPCs contain several distinct FG domains, each comprising variable repeats. Nevertheless, we now found that sequence heterogeneity is no fundamental requirement for barrier function. Instead, we succeeded in engineering a perfectly repeated 12mer GLFG peptide that self-assembles into a barrier of exquisite transport selectivity and fast transport kinetics. This barrier recapitulates RanGTPase-controlled importin- and exportin-mediated cargo transport and thus represents an ultimately simplified experimental model system. An alternative proline-free sequence forms an amyloid FG phase. Finally, we discovered that FG phases stain bright with 'DNA-specific' DAPI/ Hoechst probes, and that such dyes allow for a photo-induced block of nuclear transport.

[1] Department of Cellular Logistics, Max Planck Institute for Biophysical Chemistry, Göttingen, Germany. ✉email: goerlich@mpibpc.mpg.de

Nuclear pore complexes (NPCs) perforate the nuclear envelope and enable nucleocytoplasmic exchange through a 70-nm wide central channel[1–4]. Passage through this central channel is controlled by a sieve-like permeability barrier that grants rapid passage to small molecules but becomes increasingly restrictive as the size of the mobile species approaches or exceeds a limit of ≈30 kDa[5]. Free mCherry (≈30 kDa), for example, equilibrates between the nucleus and cytoplasm with a half time in the order of 10 min[6]. This is ≈1000 times slower than expected for a central channel without a barrier[6,7]. The actual passage rate depends, however, not just on the size of the mobile species but also on its surface properties, with exposed hydrophobic and arginine side chains speeding up NPC passage[6]. Thus, there is a continuum between passive retention by the barrier and facilitated translocation through it.

The NPC-barrier suppresses an intermixing of nuclear and cytoplasmic macromolecular contents. At the same time, it allows active, directed transport by dedicated carriers called nuclear transport receptors or NTRs for short[8,9]. NTRs carry cargoes across the pore and are optimized for facilitated translocation. Their facilitated transport is indeed remarkably efficient with a single NPC allowing nearly 1000 translocation events per second[7] and transit times as short as 10 milliseconds[10–13].

The importin β superfamily has ≈20 human members and constitutes the largest NTR class[14,15]. It comprises mediators of import into nuclei (importins) as well as exportins. Examples include transportin that imports numerous RNA-binding proteins[16], importin β itself that mediates classical NLS-mediated import together with importin α[17,18], or Exportin 1/CRM1[15,19] that exports ≈1000 different proteins[20] as well as newly assembled ribosomal subunits and other RNA-protein complexes from cell nuclei.

Importins and exportins are capable of active transport against concentration gradients, drawing energy from the nucleocytoplasmic RanGTP gradient[8]. Importins capture cargo at low RanGTP-levels in the cytoplasm, translocate through NPCs, release their cargo when encountering RanGTP inside nuclei, and return as importin·RanGTP complexes to the cytoplasm. In the cytoplasm, the RanGTPase-activating protein, RanGAP, and its co-activator RanBP1/RanBP2 then trigger GTP hydrolysis and release of RanGDP, allowing the importin to bind and import its next cargo. Exportins function similarly but recruit cargo along with RanGTP inside nuclei and (upon GTP hydrolysis) release cargo and RanGDP into the cytoplasm. Nuclear Ran is then replenished by NTF2-mediated nuclear import of RanGDP[21], followed by RanGEF-mediated nucleotide exchange to RanGTP[22]. As NTF2 has to import one Ran molecule for every importin- and exportin cycle, it probably accomplishes more transport events than any other NTR.

The so-called FG-repeat domains[23,24] are key for transport selectivity[25,26]. They are anchored to the NPC scaffold[4], are intrinsically disordered, and typically comprise a few hundred residues, including numerous FG (Phe-Gly) motifs that interact with translocating NTRs[27,28]. FG domains can also engage in cohesive, multivalent self-interactions that maintain the permeability barrier and result in sieve-like FG hydrogel structures[29–31]. FG hydrogels reconstituted from purified FG domains display permeability properties that closely resemble those of NPCs, provided that the local FG domain concentration exceeds a threshold of about 150–200 mg/ml[30]. The sieve effect of such dense FG phases excludes "inert" macromolecules, yet the "FG-philic" NTRs can rapidly enter and traverse them[6,30,32].

Yeast and vertebrate NPCs contain ≈10 different FG domains, which are anchored to distinct sites on the NPC scaffold and differ widely in prevailing FG motifs (e.g., GLFG, FSFG, or SLFG), FG motif density, and types of inter-FG spacers[33–38].

These differences in FG domain sequence and composition can profoundly impact their biophysical properties. The regular FG sub-domain (residues 274–601) of *S. cerevisiae* Nsp1[23], for example, features highly charged spacers that confer high water solubility and counteract cohesive interactions[29,31,39]. The N-terminal Nsp1 FG domain (~residues 1–175), in contrast, is charge-depleted, highly cohesive, NQ-rich, and prone to amyloid-like interactions. Some animal FG domains are extensively O-GlcNAc modified[40,41], which increases water solubility and attenuates otherwise extremely strong cohesive interactions[34].

The rather long (≈500 residues) FG domain of Nup98[27,42] appears particularly critical for the NPC barrier[43] and is special in several respects[32]: (i) it shows a strong sequence conservation (70% identity between fish and human) that is highly unusual for an intrinsically disordered protein domain[44]; (ii) it has the highest FG-density (1 motif per ≈12 residues); (iii) it is extremely depleted of charged residues, and (iv) as a consequence, it experiences water as a "poor solvent" and phase separates readily from dilute (≈1 μM) solutions to self-assemble dense FG phases of exquisite transport selectivity[32]. These properties are conserved across all eukaryotic clades and are thus probably fundamental for function.

A detailed physicochemical and structural analysis of the Nup98 FG phase systems is vital for understanding nuclear transport selectivity. However, substantial progress has been hampered by any given Nup98 FG-repeat domain being irregular along its sequence, with variable FG motifs, inter-FG distances, and inter-FG spacers. This heterogeneity makes it, for example, difficult to pinpoint which repeat units are actually cohesive. Furthermore, it is a combinatorial challenge to track down the structural principles of barrier formation when there are thousands of possibilities to combine the ≈50 different repeat units into cohesive clusters.

Starting from the *Tetrahymena thermophila* MacNup98 FG domain[32,45], we now have simplified the sequence stepwise down to a perfectly repeated GLFG-repeat domain. It comprises 52 identical G**GLFG**GNTQPAT repeat units of 12 residues length (prf.GLFG$_{52x12}$) and represents the closest possible match to the original sequence in terms of hydrophobicity, amino acid composition, as well as in dipeptide frequencies. This variant readily assembled an FG phase that faithfully recapitulates the selectivity and transport kinetics typically observed in NPCs. Moreover, by reconstituting nuclear import and export, we showed that this FG phase accurately reflects the RanGTP-dependence of active cargo transport. This suggests that FG-repeat sequence diversity is not fundamental for function. In addition, the perfectly repeated GLFG-repeat domain provides an ideal starting point for analyzing structural principles of barrier formation and sequence–function relationships. Related to the latter, we observed that eliminating proline rendered the domain highly prone to amyloid formation. We further observed that FG phases attract a diverse range of chemical fluorophores, that cohesively interacting FG domains stain very bright with "DNA-specific" dyes like DAPI or Hoechst 34580 (with rather red-shifted emissions) and that transport through NPCs gets blocked by UV illumination in the presence of DAPI or Hoechst.

## Results

**Regularization of a Nup98 FG-repeat domain to a perfectly repeated one.** With this study, we have aimed to establish the simplest possible cohesive FG system with authentic phase behavior and NPC-like transport selectivity. We chose the FG domain of the macronuclear MacNup98A from *Tetrahymena thermophila*[45] (Mac98A FG) as a starting point, because it is already well-characterized[6,32], and it is not complicated by an NQ-

rich sequence of high amyloid-forming propensity (like fungal Nup98/Nup100/Nup116 FG domains), or by obligatory O-GlcNAc modification (like its animal counterparts). It comprises a total of 666 residues: an N-terminal FG sub-domain, a Gle2-binding (GLEBS) domain[46,47] followed by a C-terminal FG sub-domain.

The FG part is of low sequence complexity and depleted of the amino acids D, E, R, H, K, C, Y, W, M, I, and V. It is dominated by GLFG motifs that appear 27 times in the sequence. However, there are also >20 FG and FG-like motifs, such as GIFG, GLLG, GMLG, and shorter FG, FS, or LG motifs, as well as isolated hydrophobic residues within the inter-FG spacers (Fig. 1a and Supplementary Table 1). Moreover, sequences and distances between FG motifs vary—with the result being that not a single repeat unit appears twice in the domain.

In order to simplify this sequence, we initially left the GLEBS domain unchanged, kept the amino acid composition as close as possible to the original (since the charge-deficient amino acid composition is likely fundamental), and implemented sequence changes in four subsequent steps leading to the variants shown in Fig. 1a (see Supplementary Note 1 for complete sequences as well as Supplementary Table 1 for amino acid composition and FG motif statistics). During steps one and two, FG-like and GLFG-like motifs were converted into GLFG motifs by conservative exchanges of hydrophobic residues (M, I, L, V→F or M, I, V→L) as well as by shifting hydrophobic residues (from the spacers to FG/ FG-like motifs). During step three, inter-FG spacers were adjusted to identical lengths—also by shifting residues (between spacers). The result was "GLFG$_{52x12}$"—an FG domain comprising 52 GLFG repeats of uniform unit length (12 residues) but still diverse spacer sequences.

The fourth step was to convert the inter-FG spacers to all identical sequences, with the boundary condition of keeping the amino acid composition close to the original sequence. Some deviations were inevitable because any amino acid used for the perfect repeats has to occur at least once per repeat unit and then account for multiples of 1/12 (≈8.3%) of the molar fraction. While the GLFG$_{52x12}$ contents of F, L, Q (1 per repeat each), and G (4.1 per repeat) matched this scheme well (see Supplementary Table 1), there were larger deviations from whole numbers for N (1.26 per repeat), A (1.27 per repeat), T (1.56 per repeat), and P (0.58 per repeat). The still closest match was to round N and A down to 1 per repeat unit, T up to 2, and P up to 1. The order of residues was set by matching dipeptide- and three-peptide frequencies to the original MacNup98A FG domain. We refer to the now perfectly repeated GG**GLFG**GNTQPAT sequence as "prf.GLFG$_{52x12}$[+GLEBS]".

For each step, we constructed codon-optimized bacterial expression vectors, expressed and purified the corresponding variants, and tested them for FG phase assembly as well as for transport selectivity (Fig. 1b). It was striking to see that the four simplified FG domains behaved essentially like the parental wild-type Mac98A FG domain: all of them phase-separated already during expression in *E. coli*, forming insoluble but transport-selective material. Following purification and dilution out of 4 M guanidinium hydrochloride, they all assembled into near-spherical "FG particles". Furthermore, all assembled FG phases excluded the inert probe mCherry very well, namely to a partition coefficient of <0.05, but at the same time allowed efficient entry of the 30-kDa nuclear transport receptor NTF2 to partition coefficients ranging between 2600 and 2900. These virtually identical numbers suggest that the perfectly repeated variant is indeed an excellent representative of the wild-type sequence.

It might appear surprising that the functional differences between the wild-type Mac98A FG domain and prf.GLFG$_{52x12}$[+ GLEBS] are so small—given that the GLFG motif count increased nearly twofold from 27 to 52. However, a straightforward explanation is that the GLFG-like, FG-like motifs, and scattered hydrophobic residues in the wild-type sequence make similar contributions to cohesive interactions and NTR-binding as the canonical GLFG motifs.

**A proline-free, perfectly repeated GLFG-repeat variant forms thioflavin-positive amyloids**. Another interpretation of the just described results is that the nature of the inter-FG spacers is not important. Such overgeneralization is, however, invalid because we had applied constraints to the sequence, namely, to keep the average spacer length, the overall hydrophobicity, and the compositional bias, including a lack of charged residues.

In fact, we had designed another perfectly repeated sequence that illustrates quite strikingly that spacer details do matter and that the above-described solution was not a trivial one. This alternative sequence comprises all-identical GG**GLFG**GATNSQT repeats. The initial content of 0.58 prolines per repeat was here rounded down to 0, and serine was included instead. We, therefore, refer to this variant as the Pro-free_prf.GLFG$_{52x12}$ (Fig. 2 and Supplementary Fig. 1).

The proline-free variant turned out to be special in that it formed fibrous FG phases that stained brightly with Thioflavin-T, which indicates amyloid cross-β-sheets[48,49]. Such behavior was previously reported for the *S. cerevisiae* Nup100 and Nup116 FG phases[32] as well as for the N-terminal Nsp1 FG domain[39], which all have a very high N/Q content of >30% and thus resemble NQ-rich prions like Sup35[50]. The N/Q content of the Pro-free_prf. GLFG$_{52x12}$ is, however, even lower (16.6%) than that of the original Mac98A FG domain (18.9%). Nevertheless, its Thioflavin-T signal was 300 times stronger and even 6 times stronger than that of the Nup116 FG domain, which had served as our positive control (see 30-min timepoints in Fig. 2c). Perhaps related to this, we observed that higher guanidinium hydrochloride concentrations were required to solubilize the Pro-free variant (6 M) compared to the wild-type MacNup98A or any of the above-described simplified FG versions (4 M).

Amyloid formation typically requires seeding and proceeds only slowly[51–54]. In the case of the N/Q-rich Nup116 FG domain, we observed that the Thioflavin-T signal accumulated with a half time of >10 h, indicating that slow (inter-) molecular re-arrangements follow the very rapid phase-separation process. In contrast, the half-maximum Thioflavin-T signal of the Pro-free_prf.GLFG$_{52x12}$ variant was reached already after ≈10 min (Supplementary Fig. 1). Thus, by chance, we had designed an FG-repeat variant with an extraordinary amyloid-forming propensity.

The lack of β-sheet-breaking prolines is a plausible but insufficient explanation of the amyloid phenotype since the Thioflavin signal hardly increased when prolines were eliminated from the original Mac98A FG domain (Supplementary Fig. 2). We, therefore, assume that also the perfectly repetitive nature of the Pro-free_prf.GLFG$_{52x12}$ contributed crucially.

**Virtually all chemical fluorophores interact with FG phases**. Given that Thioflavin-T discriminated so distinctly between different types of FG phases, we made a detour into exploring how other dyes interact with FG systems. We tested a diverse panel of 27 fluorophores (ranging from rhodamine to DAPI) and observed that all of them stained FG phases—at least to some extend (Figs. 3a and 4a; Table 1). Explanations are that these dyes are conjugated π electron systems with a potential of stacking with phenylalanine side chains of the FG phase and that their hydrophobic nature drives the association with the FG phase. This mirrors our earlier observations that exposing aromatic/hydrophobic amino acid side chains on the surface of GFP also enhances FG interactions[6].

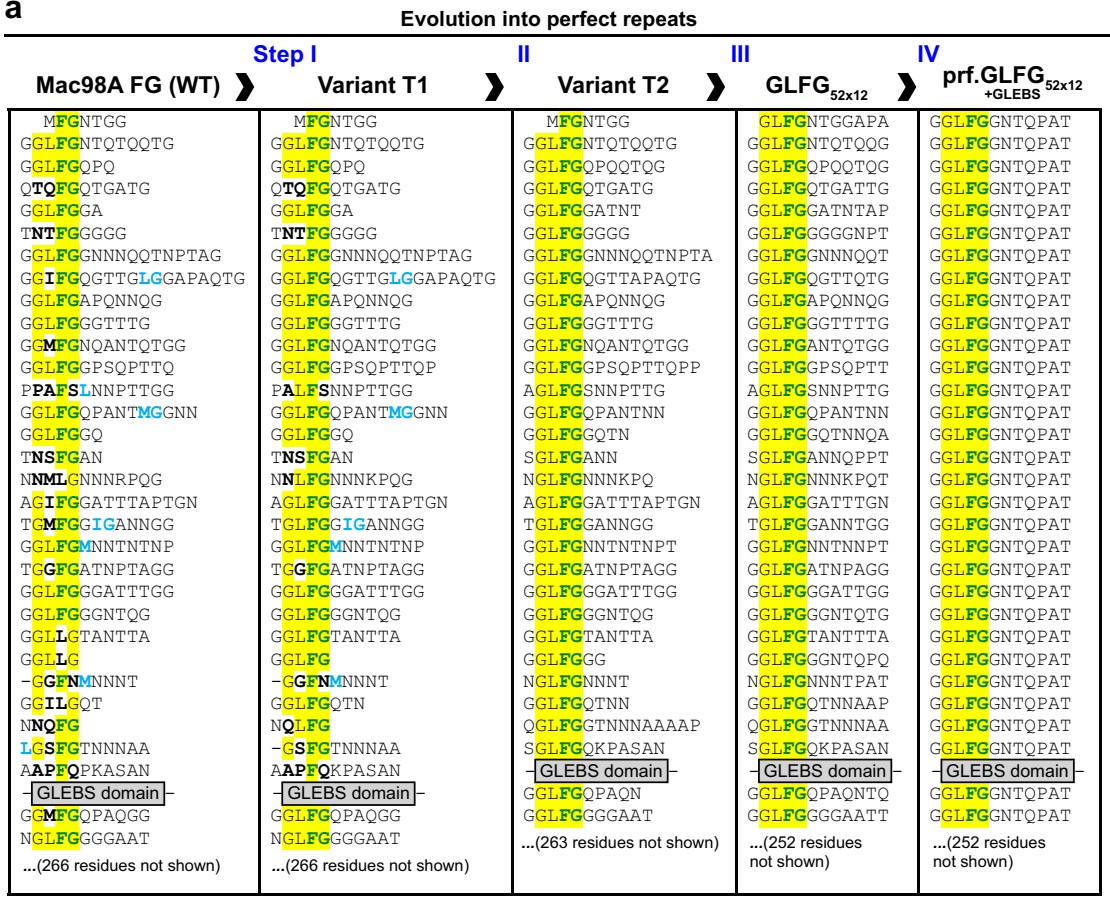

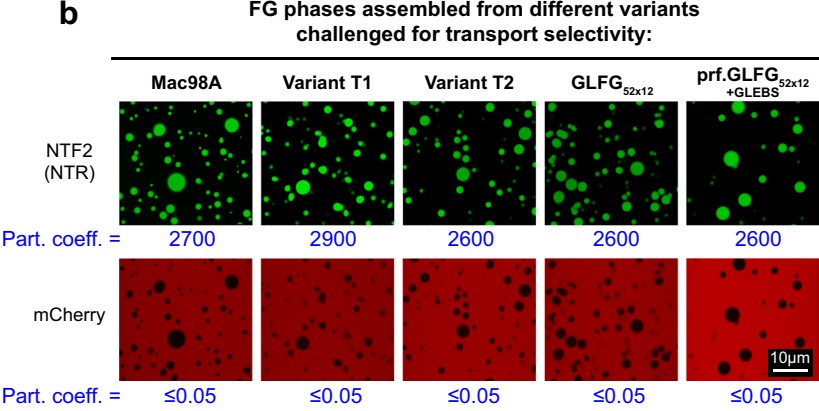

**Fig. 1 Engineering of a perfectly repeated GLFG domain with authentic phase behavior and NPC-typical selectivity. a** Sequences of the *Tetrahymena thermophila* MacNup98A ("Mac98A") FG domain and simplified variants. Mac98A contains an intervening 44-residue GLEBS domain (binding site for the mRNA export mediator Gle2p) that was kept unchanged in these variants. For space economy, only the N-terminal ≈400 residues up to the GLEBS domain are shown. The C-terminal sequences of the FG domains (≈270 residues not shown) follow the same design principle (see Supplementary Note 1 for complete sequences and Supplementary Table 1 for amino acid compositions). **b** Indicated FG domains were dissolved at a concentration of 1 mM in 4 M guanidinium hydrochloride, and phase separation was initiated by a rapid 50-fold dilution with assay buffer (50 mM Tris/HCl pH 7.5 50 mM, 150 mM NaCl, 5 mM DTT), followed by another fourfold dilution in buffer +6 μM mCherry and 1 μM Alexa Fluor 488-labeled rat NTF2. Samples were analyzed by confocal laser-scanning microscopy (CLSM). All FG domain variants phase-separated to μm-sized, spherical FG particles that excluded mCherry (red) very well and accumulated the NTR NTF2 (green) to very high partition coefficients. Note that phase separation occurred here already at 5 μM FG domain concentration, which is at least 100 times lower than the most conservative estimate for the local Nup98 FG domain concentration at NPCs (48 copies in a cylinder of 70 nm diameter and 40 nm height).

**Chemical fluorophores can strongly bias partition experiments.** There were, however, substantial differences between individual fluorophores. Cysteine-quenched Atto488 maleimide showed the weakest FG phase interaction. The ratio of inside-to-outside-FG phase fluorescence (Mac98A FG in:out) remained here as low as 2 (Fig. 3a), perhaps because it is "passivated" by optimally positioned negative charges (Supplementary Fig. 3). By contrast, Atto647N accumulated 60-fold, likely because its bulky

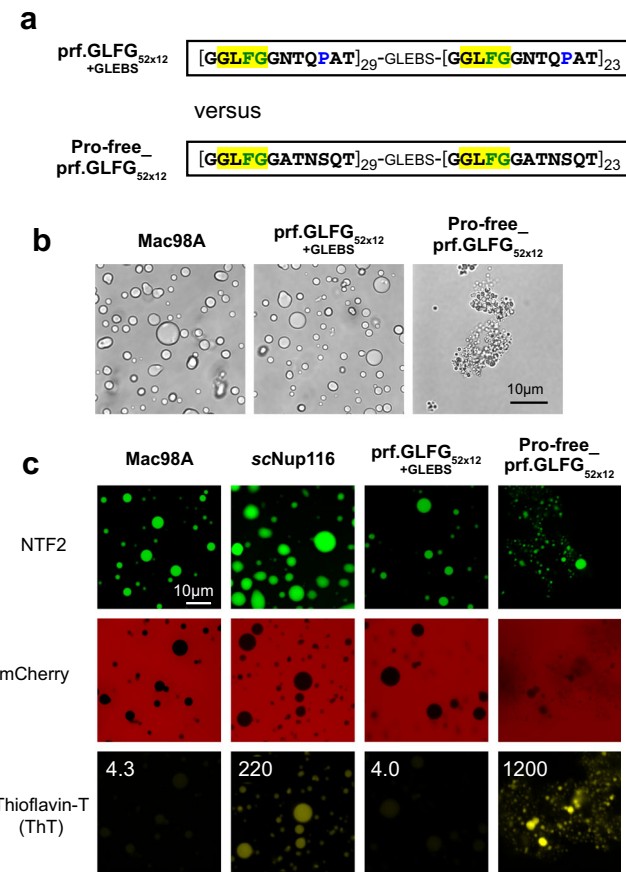

**Fig. 2 Absence of prolines in a perfectly repetitive GLFG variant favors amyloid-like structures. a** Comparison of two perfectly repetitive FG domains. The "Pro-free_prf.GLFG$_{52x12}$" is a proline-free variant (see also Supplementary Table 1). **b** Phase-contrast images of FG phases assembled from Mac98A FG domain, the proline-containing prf.GLFG$_{52x12}$[+GLEBS] and the Pro-free_prf.GLFG$_{52x12}$. Note that the former two formed spherical particles, but the Pro-free variant formed irregular shapes. The experiment was repeated independently four times with similar results, and representative images are shown. **c** FG phases were assembled from the indicated FG domains and challenged with Alexa488-labeled NTF2, mCherry, and Thioflavin-T (ThT). Numbers refer to the ratios of fluorescence inside the FG phases to that in the surrounding buffer. Images were taken at 30 min after FG phase assembly. Note that the bright ThT stain of the Pro-free variant is diagnostic of amyloid structures. The N/Q-rich Nup116 FG repeats served as a ThT positive control. See also Supplementary Figs. 1 and 2.

hydrophobic structure lacks (negatively charged) FG-repellent moieties. Cy3 even reached a partition coefficient of 110 in the Mac98A and 610 in the Nup116 FG phase (Table 1). The Nup116 FG phase was generally a stronger attractant for fluorophores (Table 1), perhaps because of its higher protein content (300 mg/ml instead of 150 mg/ml[32]). However, it seems that the sequence context and possibly the structure of FG clusters also contribute to the very strong binding of some of the dyes. Note that the scales for fluorophore-stickiness to FG phases (Table 1) and lipid bilayers[55] are quite different.

Fluorophore labeling is commonly used for tracing macromolecules in biochemical or cellular systems—with the implicit assumption that the tracing does not alter the molecules' behavior. We tried to challenge this, using the rather "FG-neutral" efGFP8Q variant[6] that enters the Nup116 phase with a partition coefficient of ≈0.8 (Fig. 3b). It was quite striking to see that attaching a single Atto647N moiety increased this number

≈30-fold; i.e., the sticky fluorophore literally "dragged" the GFP into the FG phase. The magnitude of the effect is remarkable, considering that the GFP is 35 times larger than the dye. It illustrates that chemical fluorophores can introduce a strong experimental bias when studying the partitioning of reporters into phase-separated systems. To reduce bias, one should resort to the least sticky fluorophores. Table 1 can guide such choices. Alternatively, one can use GFP derivatives whose fluorochrome is fully shielded and whose protein surface can even be tuned to be attractive or repellent to specific protein structures[6].

**Cohesive FG phases stain bright with "DNA-specific" DAPI or Hoechst dyes.** We also tested a range of environmentally sensitive dyes. These included SYPRO orange, whose fluorescence is quenched in aqueous buffers and enhanced when bound to aromatic hydrophobic amino acid side chains. This effect is commonly exploited in thermal stability experiments to monitor the unfolding of proteins[56]. SYPRO orange gave a bright FG phase-specific signal, reaching an FG in:out ratio of ≈1000 (Fig. 4a). This can be explained by FG domains being intrinsically disordered and thus exposing Phe side chains already in their functional state.

However, the most brilliant FG phase signals were obtained with DAPI, Hoechst 33342, Hoechst 33258, and Hoechst 34580, which reached Mac98A FG in:out ratios from 1700 in the case of DAPI to >10,000 in the case of Hoechst 34580 (Fig. 4a). Given that these dyes are so widely used as DNA-specific probes[56,57], this result was indeed surprising.

The bright FG phase staining in the microscopic images indicates a selective and strong binding of these dyes. Bulk measurements documented, however, also an enormous fluorescence enhancement: the Hoechst 33342 fluorescence, for example, was increased ≈160-fold by the Nup116 FG domain, ≈60-fold by the Mac98A FG domain, and ≈80-fold by the regular GLFG$_{52x12}$ repeats (Fig. 4b and Supplementary Table 2). This parallels the greatly enhanced fluorescence when DAPI or the Hoechst dyes bind DNA[58].

While these DNA dyes dock into the minor groove of AT-rich double-stranded DNA[59], we assume that they engage in π-π stacking/hydrophobic interactions with FG motifs. Such differences in binding mode are indeed evident from ligand-specific differences in the fluorescence emission spectra (Fig. 4c, d). The emission peak of Nup116 FG domain-bound Hoechst 34580, for example, is >50 nm red-shifted as compared to the corresponding DNA complex.

The Hoechst-FG phase complexes do not comply to Kasha's rule[60]: their emission peaks shift by up to 45 nm when moving the excitation wavelength from the peak to the red edge of the absorbance spectrum (Fig. 4e and see Supplementary Fig. 4 for details). Such "Red edge effect" indicates heterogeneity in the environment of the fluorescent species[61,62], which may arise from various modes of dye–FG domain contacts. There are also subtle differences in the emission spectra between different FG domains, perhaps reflecting heterogeneity in FG motifs and FG contexts.

Bulk fluorescence measurements allowed us to study dye–FG interactions independently of any phase separation (Fig. 4b and Supplementary Table 2). This revealed that the enhancement of, e.g., Hoechst 33342 fluorescence by the non-cohesive Nsp1 FG domain (residues 274–601) is minimal (twofold) as compared to the (cohesive) Nup116 FG domain (160-fold). This suggests that the Hoechst stains are rather selective for cohesive FG domains. These FG domains phase separate not only to high local FG concentration but also might form micelle-like FG clusters that are highly effective in shielding the dye against quenching by solvent molecules. This would parallel an earlier report of

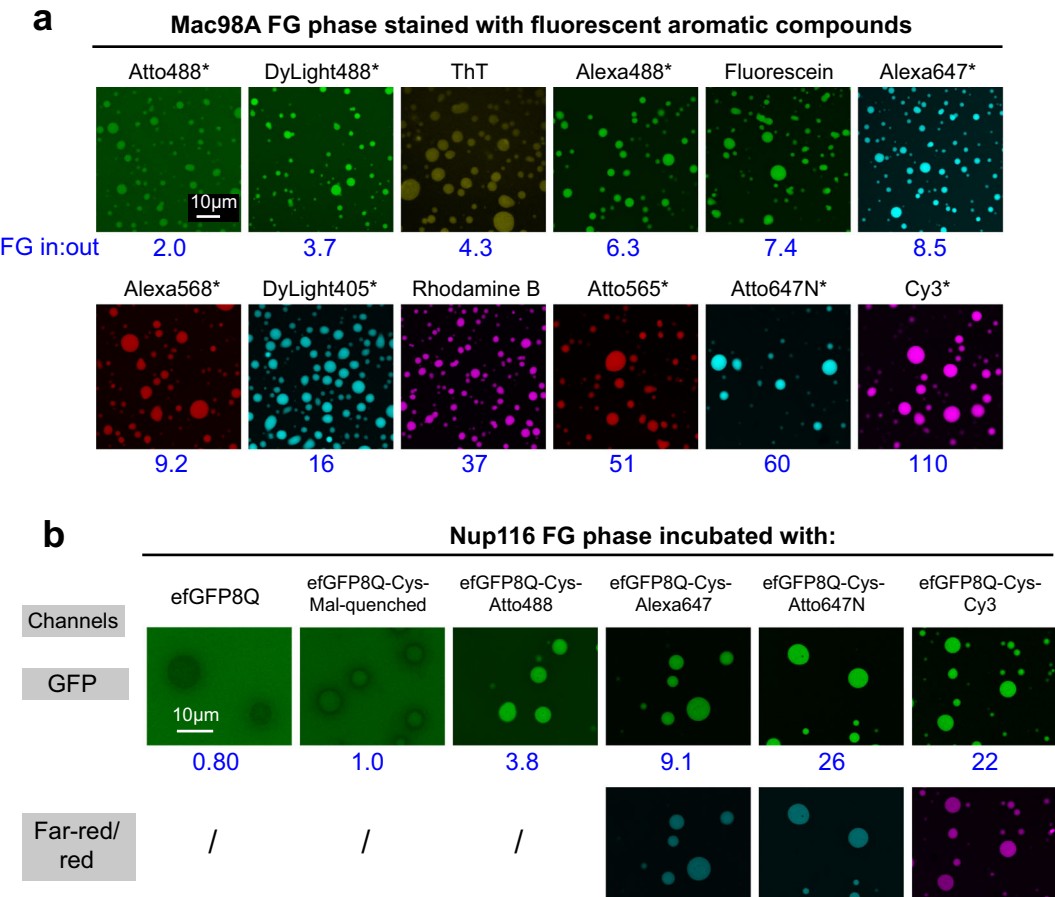

**Fig. 3 FG phases stained by fluorescent aromatic compounds. a** Mac98A FG particles were formed, incubated with 2 μM of indicated fluorophores, and analyzed by CLSM. Note that all fluorophores got attracted by the phase; however, there were great differences with Atto488 being nearly inert and Cy3 accumulating very strongly. Table 1 lists all fluorophores tested and their partition coefficients. Chemical structures are shown in Supplementary Fig. 3. * indicates that a cysteine-quenched fluorophore maleimide was tested. Blue numbers: ratios of signals inside Mac98A FG phase to signal in the surrounding buffer. Scan settings/image brightness were adjusted individually. **b** scNup116 FG particles were challenged with efGFP8Q, or variants with an added C-terminal cysteine that had been modified either with maleimidopropionic acid ("Mal-quenched"), Atto488-, Alexa647-, Atto647N-, and Cy3-maleimides, respectively. Blue numbers: ratios of GFP fluorescence inside: outside the Nup116 FG phase. The far-red channel detects signals from Alexa647/Atto647N (colored cyan), and the red channel detects signals from Cy3 (colored magenta). Scanning settings/image brightness were adjusted individually. Note that, e.g., the Atto647N modification increased the GFP partition coefficient ≈30-fold.

Hoechst 3342 being a selective probe for micellar forms of detergents[63].

Notably, DAPI or the Hoechst dyes had no obvious effect on phase separation or the partition selectivity of the analyzed FG phases (Fig. 4f). Furthermore, these dyes can conveniently be excited by the 405-nm laser line and spectrally well separated from GFP, mCherry, and far-red signals. Also, they labeled all condensed FG phases tested so far (see, e.g., Fig. 4g). Therefore, we expect them to become useful tools for imaging and analyzing in vitro assembled FG phases—without the need for covalently modifying the corresponding FG domain(s). In addition, the significant red edge excitation shift may be a sensitive reporter for dynamics of these condensates[62].

**A UV light-induced block of NPCs.** Digitonin-permeabilized cells with leaky plasma membranes and intact nuclear envelopes are a powerful system for studying transport through NPCs[7,64] because it allows to introduce fluorescent transport substrates into these cells and to follow and quantify their transport between the cytoplasm and cell nucleus. When such an experiment is performed live, it is necessary to pre-focus the samples on the stage of a confocal fluorescence microscope.

When using DAPI or Hoechst 33342 and higher energy UV light illumination at 365 nm (from a mercury lamp passed through a bandpass 360/40-nm excitation filter) for this purpose, we noticed a striking effect: NPCs of illuminated HeLa nuclei became blocked for importin β-mediated import of IBB-GFP fusions (Fig. 5). The effect was specific as it was dye-dependent, restricted to illuminated nuclei, and not observed when illumination was changed to 405 nm. The chemical nature of the effect is still unclear, but a light- and dye-induced crosslinking between FG domains would be the simplest explanation, analogous to the inhibitory action of wheat germ agglutinin[5,65]. We do not know which FG domain(s) or which kind of FG domains/motifs constitute the immediate target. The prospect of switching NPCs optically, however, opens exciting experimental opportunities.

**A GLEBS-free, perfectly repeated FG variant.** The GLEBS domain still interrupts the prf.GLFG$_{52x12}$[+GLEBS] domain. Its deletion led to "prf.GLFG$_{52x12}$", composed solely of 52 consecutive identical 12mer GLFG repeats (Fig. 6a). This now ultimately simplified FG domain still assembled an FG phase of exquisite transport selectivity—with partition coefficients

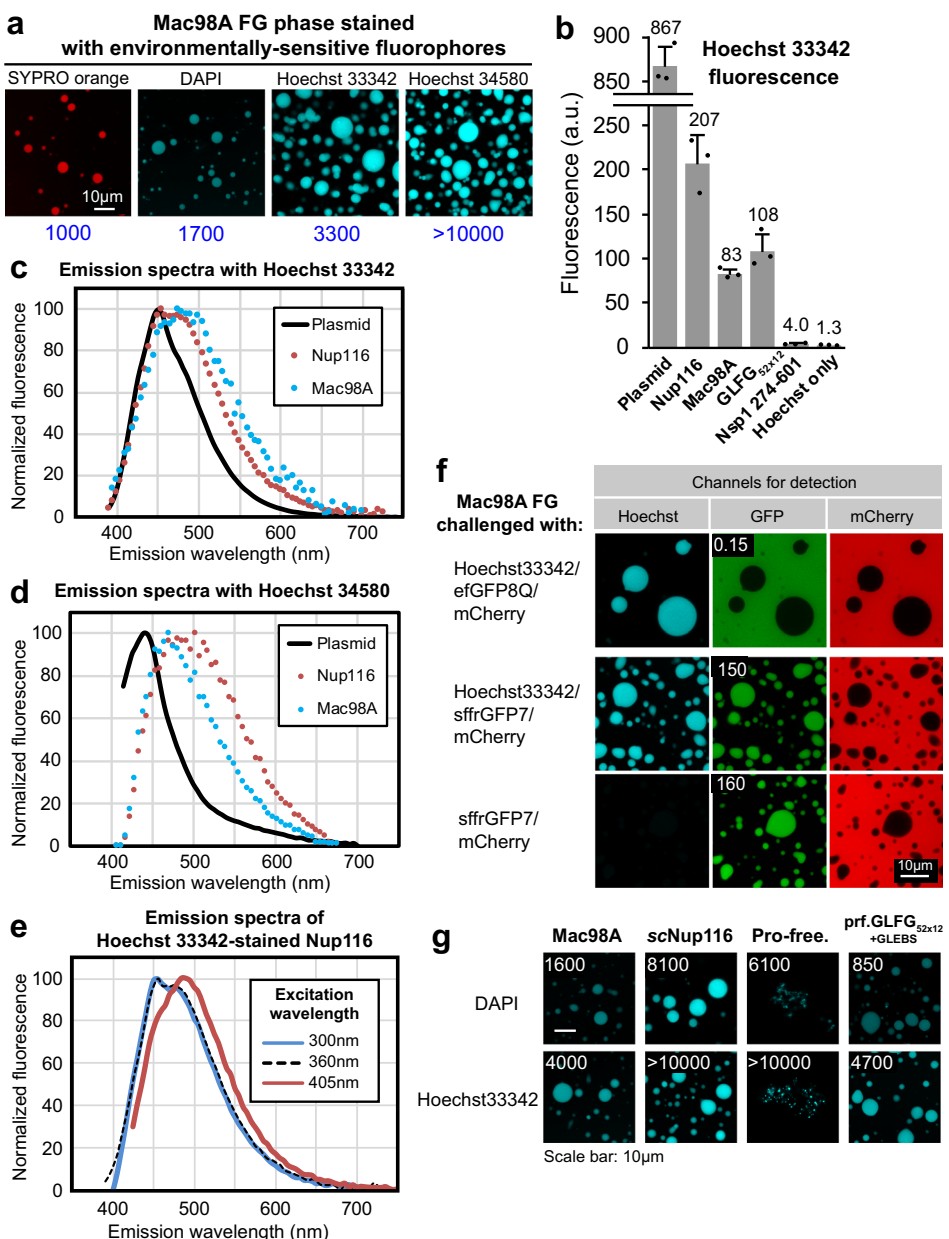

**Fig. 4 Bright staining of condensed FG phases with "DNA-specific" DAPI and Hoechst dyes. a** Environmentally sensitive fluorophores (SYPRO orange, DAPI, and Hoechst dyes) were used to stain Mac98A FG particles. Note the extremely high fluorescence in:out ratios (blue numbers). **b** Bulk fluorescence of FG domains (2.67 mg/ml) stained with 20 µM Hoechst 33342. Excitation was at 360 nm, detection at 440–480 nm. Data are presented as mean values (arbitrary unit, a.u.) of three independent replicates, with individual data points shown as dots. Error bars are the standard deviations. Note that the cohesive FG domains (Nup116, Mac98A, and GLFG$_{52x12}$) gave strong signals, while the signal for non-cohesive Nsp1 FG fragment (residue 274–601) was at least 20 times weaker. Source data are provided as a Source Data file. **c** Emission spectra of Hoechst 33342-stained plasmid, Nup116 and Mac98A FG phases (excitation wavelength: 360 nm). Note the red-shifted fluorescence of the FG phases. **d** Emission spectra of Hoechst 34580-stained plasmid, Nup116 and Mac98A FG phases (excitation wavelength: 405 nm). **e** Emission spectra of Hoechst 33342-stained Nup116 FG phase recorded at three distinct excitation wavelengths: 360 nm corresponds to the absorption peak (see Supplementary Fig. 4) and excitation with a shorter wavelength (300 nm) showed essentially the same emission spectrum. However, excitation with a longer wavelength (405 nm, at "red edge" of the excitation spectrum) led to a red-shift of the emission spectrum and of the emission peak, known as the "red edge effect". Source data (**c–e**) are provided as a Source Data file. **f** Mac98A FG phase was challenged with the indicated probes and imaged by confocal microscopy. sffrGFP7 is an FG-philic GFP variant[6]. Detection wavelengths: Hoechst/410–470 nm, GFP/500–555 nm, and mCherry/577–700 nm. Note that the presence of Hoechst did not bias the partitioning of either sffrGFP7 or mCherry. **g** Different FG phases were stained with DAPI/Hoechst. Image brightness was adjusted individually. "Pro-free.": Pro-free_prf. GLFG$_{52x12}$. Numbers refer to FG in:out fluorescence ratios.

spanning a ≥ 50,000-fold range between the best excluded inert mobile species and well-accumulating NTRs (Fig. 6b).

To assess how the GLEBS-deletion and sequence simplification impacted phase behavior and selectivity, we compared FG phases

derived from four FG domain variants: Mac98A wild-type, Mac98A ΔGLEBS, prf.GLFG$_{52x12}$[+GLEBS], and the (GLEBS-free) prf.GLFG$_{52x12}$ domain. In several aspects, the four behaved remarkably similarly: they all excluded mCherry perfectly well

**Table 1 Mac98A/Nup116 FG phases stained by fluorescent aromatic compounds (complete list).**

| Environmentally insensitive fluorophores | *Tt*Mac98A<br>FG in:out | *sc*Nup116<br>FG in:out |
|---|---|---|
| Abberior STAR 635P* | 16 | N.D. |
| Alexa488* | 6.3 | 36 |
| Alexa568* | 9.2 | 11 |
| Alexa594* | 21 | 25 |
| Alexa647* | 8.5 | 13 |
| Atto488* | 2.0 | 2.0 |
| Atto565* | 51 | 47 |
| Atto647N* | 60 | 110 |
| Cy3* | 110 | 610 |
| Cy5* | 88 | 180 |
| DyLight405* | 16 | N.D. |
| DyLight488* | 3.7 | N.D. |
| Fluorescein | 7.4 | N.D. |
| Fluorescein* | 8.7 | 10.3 |
| OregonGreen488* | 12 | N.D. |
| Rhodamine B | 37 | 110 |
| Sulfo-Cy3* | 66 | 130 |
| Sulfo-Cy5* | 34 | 28 |

| Environmentally sensitive fluorophores | *Tt*Mac98A<br>FG in:out | *sc*Nup116<br>FG in:out |
|---|---|---|
| DAPI | 1700 | 8100 |
| Ethidium bromide | 22 | N.D. |
| Hoechst 33258 | 3300 | N.D. |
| Hoechst 33342 | 3300 | >10000 |
| Hoechst 34580 | >10000 | >10000 |
| Nuclear yellow | 970 | N.D. |
| Oxazole yellow | 380 | N.D. |
| SYPRO orange | 1000 | 1090 |
| Thioflavin-T | 4.3 | 220 |

*N.D.* not determined.
The fluorophores are arranged in alphabetical order. * indicates that a cysteine-quenched fluorophore maleimide was tested.

(FG in:out <0.05) and accumulated NTF2 very strongly (FG in: out = 2500–2700). All four FG phases recapitulated that surface arginines are facilitating NPC passage through polyvalent cation–π interactions, i.e., the phases accumulated the FG-philic, Arg-rich sffrGFP4 variant[6] (FG in:out = 12–40), but entirely excluded the corresponding 25 R→K version, where all arginines had been changed to lysines. Finally, all four phases showed the same, very high accumulation of the tetrameric 110 kDa GFP$^{NTR}$_3B7C variant (FG in:out = 700–2000), which had been evolved to mirror an NTR-like behavior[6].

However, there were also noteworthy differences. First, the GLEBS-deletion caused a sixfold increase in critical concentration for phase separation (to ≈1.7 μM), indicating a so far unexpected, positive contribution of the GLEBS domain to phase separation and inter-FG domain cohesion (Fig. 6c). The latter conclusion is supported by fluorescence recovery after photobleaching (FRAP) experiments (Fig. 6d, e). These indicate that the GLEBS-free Mac98A and prf.GLFG$_{52x12}$ domains have gained mobility within the phase—to an estimated diffusion coefficient of ≈0.04 μm$^2$/s. This is still ≈800 times slower than in free solution but at least 200 times faster than the corresponding GLEBS-containing versions[32]. This suggests that the GLEBS domain contributes to cohesive interactions and that highly selective FG phases can span a continuum of viscosities.

FG phases exclude inert macromolecules because of unfavorable FG interactions[6]. In the case of the Mac98A FG phase, this effect is so strong that even an NTR-assisted entry of rather small cargo, like EGFP (≈ 30 kDa), gets impeded[32]. For example, importin β can only moderately accumulate an IBB-EGFP fusion within this phase (FG in:out ≈10; Fig. 6b). However, the GLEBS-free phases (Mac98A ΔGLEBS and prf.GLFG$_{52x12}$) allowed a 5–12-fold higher accumulation (to partition coefficients of up to 120); they appear thus more akin to NPCs.

One function of the NPC barrier is to restrict the efflux of free RanGTP and thus to prevent a dissipation of the nucleocyto-

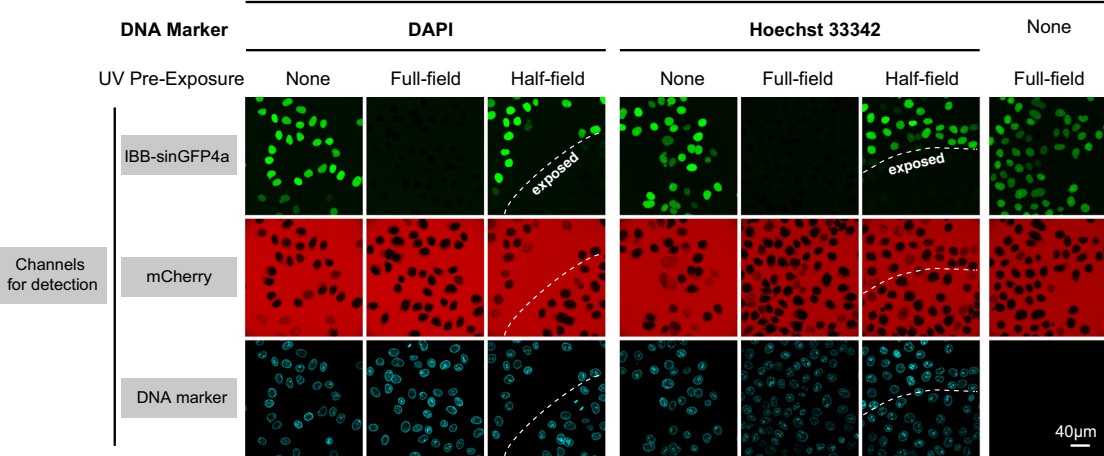

**Fig. 5 UV-induced blocking of NPCs by DAPI/ Hoechst.** HeLa cells were grown in cell culture wells, permeabilized with digitonin, and preincubated without or with the traditional DNA markers DAPI or Hoechst 33342. Indicated wells were then partially exposed for 30 s with UV light from the microscope's mercury/metal halide lamp (wavelength ≈ 365 nm). A mixture of importin β, IBB-sinGFP4a (an importin β-dependent import cargo), mCherry (as a passive exclusion marker), components of the Ran system, and an ATP/GTP-regenerating system ("Methods") was added. Import, followed live by confocal laser-scanning microscopy, was allowed to proceed at 21 °C. The micrographs show the ≈200 s time point of either a non-UV exposed area or a fully exposed one ("full-field"). "Half-field" indicates the region where the illumination boundary (dashed line) had crossed. Strong accumulation of IBB-sinGFP4a was observed in nuclei not pre-exposed to UV or samples without DAPI/Hoechst addition. The combination of DAPI or Hoechst with UV illumination blocked the import of IBB-sinGFP4a completely.

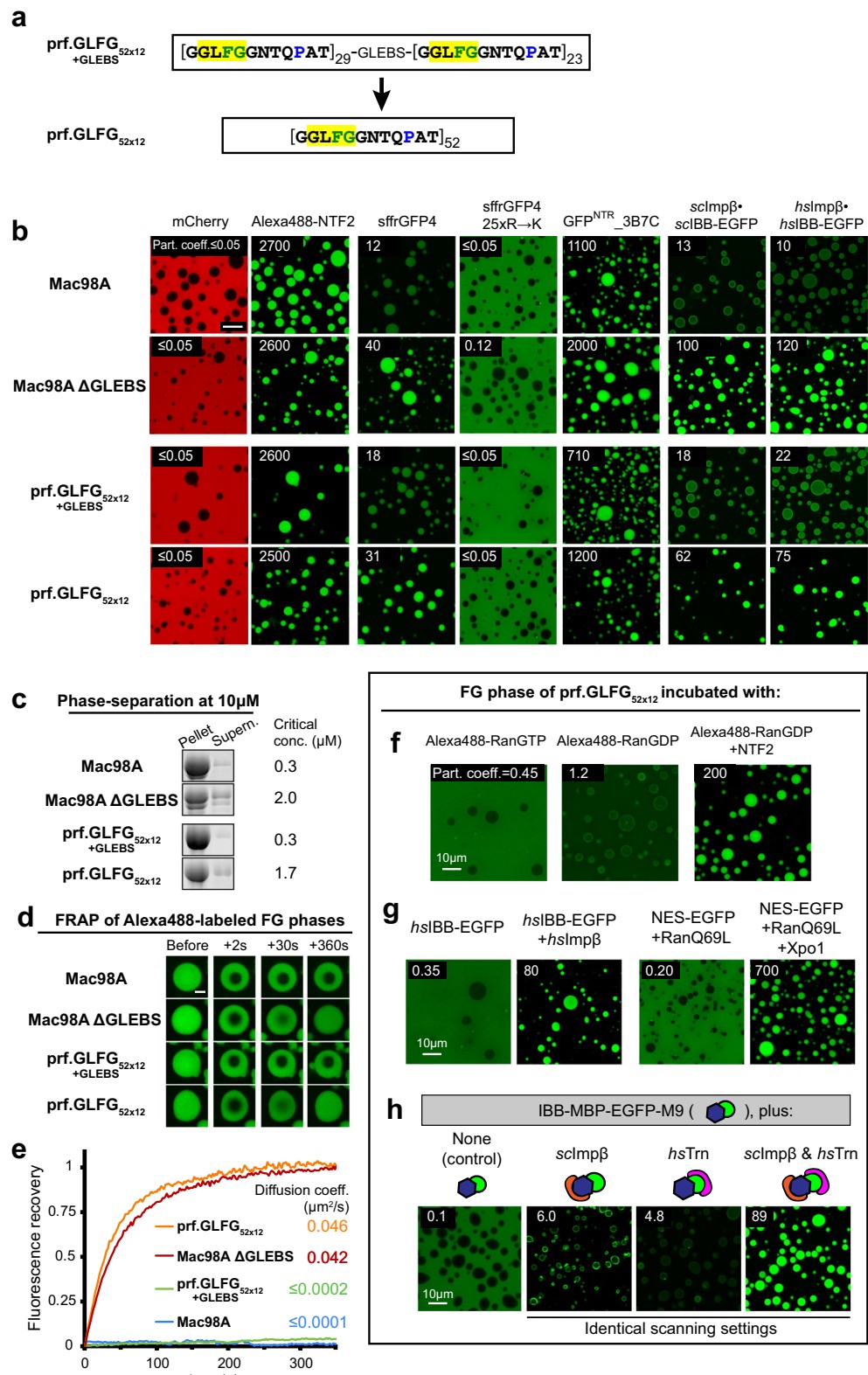

plasmic RanGTP gradient. On the other hand, the barrier must permit an efficient (NTF2-mediated) import of RanGDP. Indeed, the prf.GLFG$_{52x12}$ phase excluded RanGTP (wild-type Ran charged by RanGEF with GTP) remarkably well—given that it had to be labeled with an FG-philic fluorophore (Alexa488) for microscopic detection (Fig. 6f). RanGDP (pre-treated with RanGAP) showed a ≈ threefold higher partition coefficient with

even stronger enrichment on the FG particles' surface, indicating that the conformational switch already exposed some FG-binding sites on Ran itself. The addition of NTF2 boosted the partition coefficient of Ran by another factor of 170. This can be seen not only as a specificity control but also as a reconstitution of a crucial part of the RanGTPase cycle—with the simplest possible setup.

**Fig. 6 Selectivity of FG phases assembled from Pro-containing perfect repeats. a** An ultimately simplified FG domain (prf.GLFG$_{52 \times 12}$) was generated by deleting the GLEBS domain from prf.GLFG$_{52 \times 12}$[+GLEBS]. **b** FG phases were challenged with different protein probes for selectivity. Scanning settings/ image brightness were adjusted individually due to the large range of signals. Note that GLEBS-free FG phases (Mac98A ΔGLEBS and prf.GLFG$_{52 \times 12}$) allowed for a stronger accumulation of NTRs, NTR·cargo complexes, and the NTR-like sffrGFP4. However, the conversion of the Mac98A FG domain to perfect repeats caused only small changes in NTR-accumulation. Scale bar: 10 µm. *sc: Saccharomyces cerevisiae*; *hs: Homo sapiens*. **c** In total, 10 µM of indicated FG domain variants were allowed to phase separate and the FG phases were pelleted by ultracentrifugation. Equivalent ratios of pellets and supernatants were analyzed by SDS-PAGE/Coomassie-staining. Critical concentrations for phase separation were taken as the concentrations that remained in the supernatants ("Methods"). The experiment was repeated independently three times with similar results, and representative images are shown. Full scans of gels with molecular weight markers were provided as a Source Data file. **d** FG particles spiked with 2% Alexa488-(covalently) labeled FG domains (of the same species) were photobleached. Fluorescence recovery after photobleaching (FRAP) was recorded over time. Scale bar: 3 µm. **e** Recovery curves corresponding to (**d**). Y axis: recovery was normalized to 1 for a complete recovery. The numbers are translational diffusion coefficients derived from fitting the datasets to theoretical recovery curves ("Methods"). The GLEBS-containing Mac98A and prf.GLFG$_{52 \times 12}$[+GLEBS] FG domains were essentially immobile. Source data are provided as a Source Data file. **f** Alexa488-labeled Ran was converted either to the RanGTP form (by pre-incubation with RanGEF and an ATP/GTP-regenerating system), or to the RanGDP form (by pre-incubation with RanGAP). Panels show the partitioning of these Ran forms into the ultimately simplified prf.GLFG$_{52 \times 12}$ phase. Where indicated, unlabeled NTF2 (2 µM dimer concentration) was also added. **g** prf.GLFG$_{52 \times 12}$ phase was challenged with import (IBB-EGFP) and export cargoes (EGFP fused to a strong Xpo1-dependent nuclear export signal (NES) + RanQ69L, a mutant of Ran locked in the GTP form) in the absence or presence of the corresponding NTRs: Impβ/Xpo1. **h** prf.GLFG$_{52 \times 12}$ phase was challenged with an 80 kDa import cargo carrying two orthogonal import signals, an IBB (recognized by Impβ) and an M9 domain (recognized by transportin, Trn). Note that strong intra-phase accumulation was only observed in the presence of both NTRs, recapitulating the requirement for large cargo transport through NPCs. Numbers refer to GFP fluorescence ratios in the central regions of the particles to that in the surrounding buffer.

Figure 6g shows further specificity controls, namely that the entry of an IBB-EGFP fusion into the prf.GLFG$_{52 \times 12}$ phase is strongly (200-fold) stimulated by importin β, while Exportin 1 (Xpo1) stimulated phase entry of an NES-GFP fusion 3500-fold.

Large, hydrophilic cargoes require more than one NTR for efficient NPC passage[66,67] and FG phase entry[32]. This behavior can already be tested with an 80 -kDa MBP-GFP fusion carrying two orthogonal import signals[66], an IBB domain for importin β-dependent[68] and an M9 domain for transportin-dependent[69] import. The combination of importin β and transportin transported this IBB-MBP-GFP-M9 fusion >10 times faster through NPCs than a single NTR[66]. When tested with the prf. GLFG$_{52 \times 12}$ phase, we observed that either NTR alone conferred only a weak cargo entry (FG in:out 5–6) with intermediates accumulating at the buffer-FG boundary (Fig. 6h). Importin β and transportin together, however, allowed a robust intra-phase accumulation to a partition coefficient of 89. This emphasizes that our ultimately simplified FG domain indeed recapitulates also this aspect of NPC transport selectivity very well.

**FG phase of perfectly repeated variant allows fast intra-phase diffusion of NTRs.** Translocation through NPCs involves barrier entry and barrier passage, and we thus wondered if NTRs would move fast enough through the prf.GLFG$_{52 \times 12}$ phase to be consistent with the previously reported dwell times at NPCs[10–13]. Passage times scale inversely with the intra-phase diffusion coefficient and directly with the square of the barrier's thickness (see "Methods", Eq. 2). By FRAP, we obtained for Alexa488-labeled NTF2, 0.13 µm²/s as a diffusion coefficient in the prf. GLFG$_{52 \times 12}$ phase and 0.07 µm²/s in the +GLEBS variant (Fig. 7a, b and Supplementary Fig. 5). The exact thickness of the NPC barrier is still unknown but can be estimated. One extreme assumption would be that the Nup98 FG phase forms a separate layer. Considering a preferred concentration of 150 mg/ml[32], 48 Nup98 copies per NPCs, 60 kDa mass per domain, and 70 nm diameter for the central channel (as suggested by recent NPC structures[4,37,38,70]), this would yield a thickness of 2 nm or 20–40 µs of diffusion time (see also Supplementary Table 3). This is comfortably shorter than the 6 milliseconds reported dwell time of NTF2 at NPCs[11]. However, it needs to be considered that other FG layers then need to be passed as well. As the other FG domains are less cohesive than the Nup98 FG domain[34], one should assume faster diffusion through these other layers. An

alternative assumption would be that the ≈15-MDa FG domain mass per NPC forms a mixed FG layer. This would result in a 10-nm-thick layer and a passage time of ≈0.5–1 milliseconds.

NPC passage also includes barrier exit. To recapitulate this, we immobilized prf.GLFG$_{52 \times 12}$ particles onto glass slides, preloaded them with NTF2 to a partition coefficient of ≈2000, and washed them stepwise with fresh buffer (Fig. 7c). The washing steps left the FG tracer signal unaffected, but each step led to a drop in intra-phase NTF2 concentration. Thus, the entry into this FG phase is reversible.

**RanGTPase-controlled cargo transport in and out of the FG phase.** Importin·cargo complexes assemble in the cytoplasm before they translocate through NPCs. The binding of RanGTP to importins then triggers cargo release into the nucleus and marks an irreversible termination step in the corresponding transport cycle. In the absence of nuclear RanGTP, importin α/β/NLS or importin β·IBB complexes arrest at NPCs[71,72], suggesting that cargo release and exit from the barrier are coupled to each other.

To recapitulate this process, we immobilized prf.GLFG$_{52 \times 12}$ particles on a microscopic slide and preloaded them with an importin β·IBB-EGFP complex. Without further addition, the intra-phase signal plateaued at a partition coefficient of 77. Upon addition of RanGTP (used as a GTP-locked Q69LΔC mutant), however, the IBB-EGFP signal rapidly declined (by ≈ 90% within 10 min), indicating an enforced barrier exit (Fig. 8a–c and Supplementary Fig. 7).

We had fused Ran to a 70-kDa MBP-mCherry module, which prevents entry of Ran deep into the particles and thus confines the RanGTP-binding reaction to the surface of the FG phase (Fig. 8a). Therefore, the importin·cargo complexes had to diffuse from the interior to the FG particles' surface before getting unloaded. The overall phase-exit can (in first approximation) be fitted to a single exponential decay function (Fig. 8b, c), whereby the time constant ($k$) depended on the particle's size: the IBB-EGFP signal in smaller particles declined faster than in the larger ones (e.g., a half time of 158 s at a particle radius of 7 µm versus 64 s at a radius of 2.6 µm).

This size dependence (Fig. 8d) is plausible since smaller particles have larger surface: volume ratios and shorter diffusion distances. This consideration allows a rough estimation of timescales for barrier exit at NPCs. Suppose the diffusion through the FG particles was rate-limiting. In that case, the diffusion time

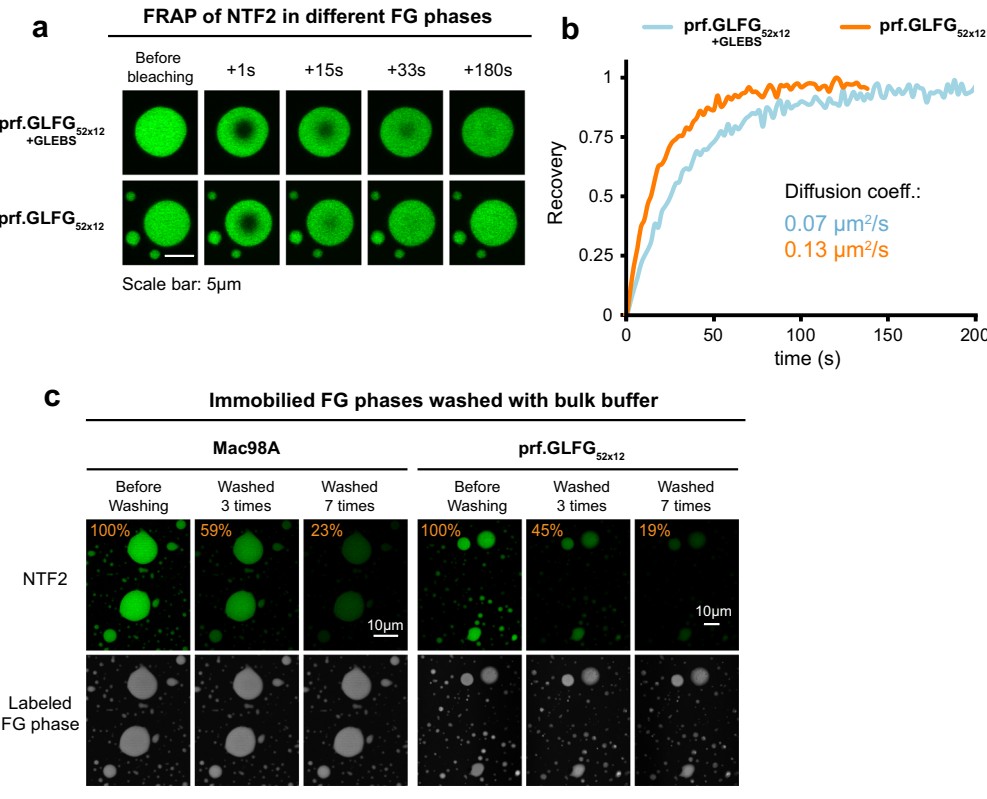

**Fig. 7 Mobility of NTRs inside FG phases assembled from Pro-containing perfect repeats. a, b** FG particles (unlabeled) assembled from both prf. GLFG$_{52x12}$ versions were preloaded with Alexa488-labeled NTF2, bleached, and FRAP was recorded over time. Experiments were repeated with multiple FG particles ($n = 5$). Representative images (**a**) and recovery curves (**b**) are shown. See Supplementary Figs. 5 and 6 for the complete dataset and other FRAP datasets for GFP-NTR derivatives. Source data are provided as a Source Data file. **c** FG particles spike with Alexa647 (covalently)-labeled FG domain molecules (FG-tracers) were immobilized on glass, preloaded with Alexa488-labeled NTF2, washed multiple times with fresh buffer ("Methods") while the fluorescence signals were recorded. Orange numbers: % Alexa488 signal inside FG phase relative to the initial signal. For both Mac98A and prf.GLFG$_{52x12}$ (GLEBS-free), signals of NTF2 within FG particles dropped to ≈20% of the original after seven rounds of washing. The majority of FG particles remained immobilized after washing, indicating their stickiness to the support.

should scale inversely with the square of the distance, i.e., we would expect typical timescales for barrier exit of 6 milliseconds for a 40-nm-thick barrier or 0.4 milliseconds for a 10-nm-thick barrier. However, it is well possible that in this case the on-rate for the RanGTP-binding to the importin becomes rate-limiting. In any case, these numbers are in line with the 10 milliseconds passage time measured for importin β-dependent transport through NPCs[12]. Non-fused RanGTP wild-type caused an approximately two times faster efflux (Supplementary Fig. 8)— consistent with its faster diffusion in such efflux experiment.

To reconstitute the exportin-mediated transport cycle, we used the aforementioned NES-GFP fusion. Without further addition, the NES-fusion remained well excluded from the prf.GLFG$_{52x12}$ phase (partition coefficient = 0.24, Fig. 9a). Xpo1 alone caused a weak accumulation inside the FG phase (partition coefficient = 5). However, with Xpo1 and GTP-loaded wild-type Ran together, the cargo accumulated to a partition coefficient of ≈700. This recapitulates that RanGTP switches the exportin to a high NES-affinity conformation[73]. The ten times stronger accumulation compared to the analogous import complex (compare Figs. 8a and 9a) indicates a particularly high "translocation power", which is in line with the fact that Xpo1 also exports huge cargoes, such as the MDa-sized 60S ribosomal subunits[74] that are harder to immerse in the FG phase than smaller ones.

In cells, Xpo1 discharges cargoes on the NPC's cytoplasmic side, where RanGAP and RanBP1/RanBP2 keep RanGTP-levels low and thus maintain a steep RanGTP gradient[75]. We tested here if such cargo release from a "pre-filled" FG phase can be

reconstituted. We found that RanGAP alone had no effect (Fig. 9b). This is consistent with earlier findings that importins and exportins bind RanGTP in a GAP-resistant manner[76]. RanBP1, however, lowered the intra-phase NES-EGFP signal ≈30-fold, probably by disassembling the Xpo1·RanGTP·cargo complex (Fig. 9e). The initially released RanBP1·RanGTP dimer is a preferred substrate of RanGAP, which triggers GTP hydrolysis[77,78] and makes the dissociation irreversible. Indeed, the combination of RanBP1 and RanGAP lowered the intra-phase NES-EGFP signal to the level of the minus-Ran control in Fig. 9a. The RanBP1-and-RanGAP-enforced export cargo exit from the FG phase occurred rapidly and was again faster for smaller FG particles (Fig. 9c, d and Supplementary Fig. 9). The observed rates would translate to 10-millisecond timescales when extrapolated to NPC dimensions. Thus, RanGTPase-dependent exit of import or export cargo from the NPC permeability barrier can be recapitulated in a minimalistic FG phase system (summarized in Figs. 8e and 9e) with realistic timescales—even when the initial partition coefficients had been very high.

## Discussion

The permeability barrier of NPCs combines a tremendous transport capacity with high selectivity. It relies on cohesively interacting FG-repeat domains that assemble into a reversibly crosslinked, hydrogel-like sieve. Mobile species smaller than the mesh size can freely pass the aqueous regions of the hydrogel. Partitioning and passage of larger ones, however, require a

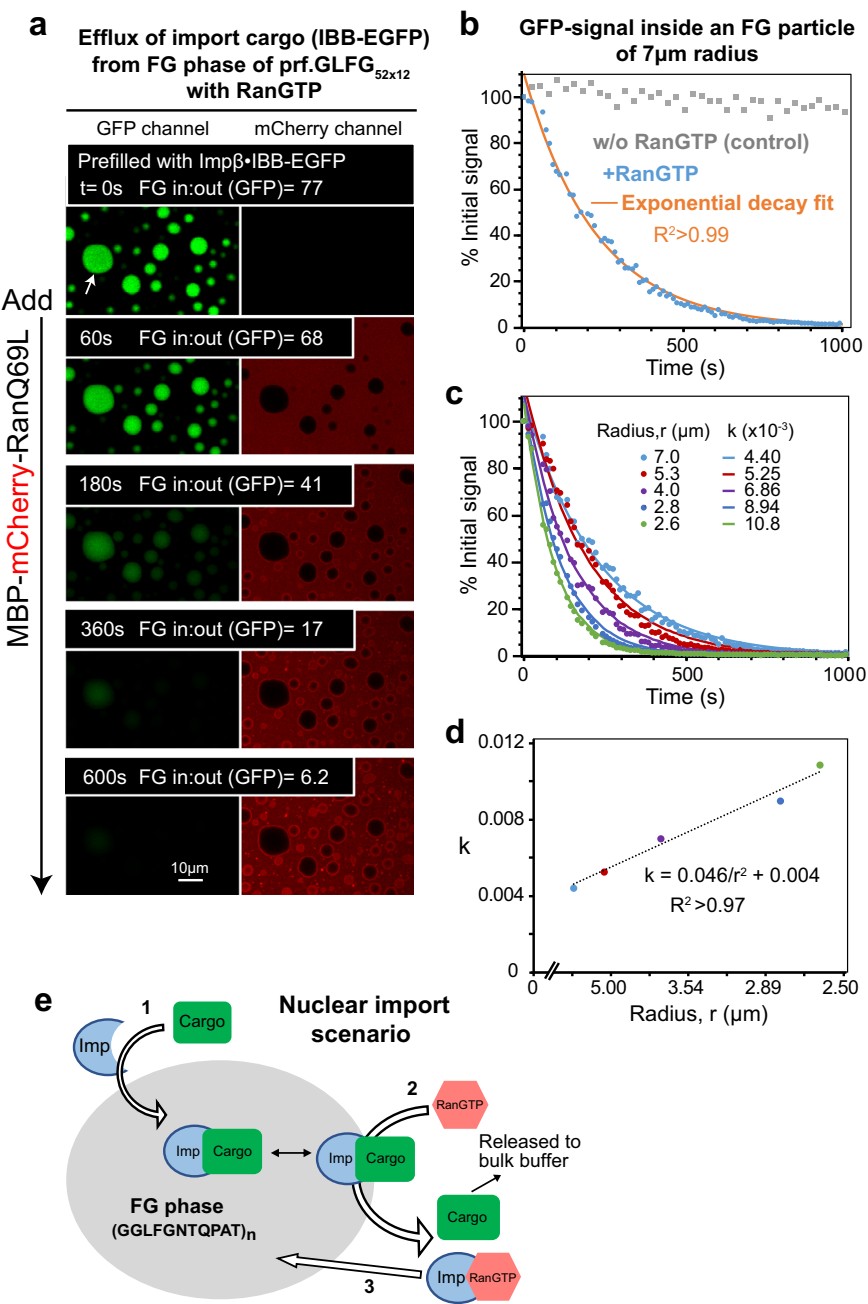

**Fig. 8 Recapitulation of RanGTPase-controlled nuclear import with the FG phase assembled by the perfectly repetitive variant. a** The FG phase of prf. GLFG$_{52x12}$ was initially loaded with $hs$Impβ·$hs$IBB-EGFP complex. At time = 0 s, a RanQ69L fragment fused to MBP-mCherry was added, which triggered the unloading of IBB-EGFP from Impβ and thus an efflux from the FG phase. Fluorescence signals of GFP and mCherry were recorded over time. The ratios of GFP fluorescence inside:outside an FG particle of 7 μm radius (marked with a white arrow) were quantified. Note that some Ran fusion was recruited to the FG phase by Impβ but only arrested at the rim of the FG particles, as expected from the FG-phobic effect of the MBP-mCherry group. This was meant to ensure that the RanGTP-reaction occurs only at the particles' surface. **b** Time course of IBB-EGFP signal inside the FG phase. Blue colored: GFP signal inside an FG particle of 7 μm radius (marked with a white arrow in (**a**)), normalized to % of the initial signal. Gray: a control set without RanGTP addition. Orange: best-fit to a single exponential decay function: $f(t) = Ae^{-kt} + B$, where $t$ is the time, $A = 110$, $B = 0$, $k = 4.4 \times 10^{-3}$. **c** The GFP signals inside FG particles with different radii were fitted to the single exponential decay function to obtain the respective time constants $k$. Source data are provided as a Source Data file. **d** $k$ obtained in the experiment described above was plotted against $1/r^2$, where $r$ is the FG particle's radius. **e** Illustration of the above processes.

transient interruption of cohesive FG-FG interactions. This comes with energetic costs that NTRs compensate for by favorable FG·NTR interactions, while inert molecules with FG-phobic surfaces remain excluded and pass NPCs only slowly.

Any more profound understanding of the sorting process would require structural insights into the nature of the barrier-relevant cohesive interactions, which are, however, still only

fragmentary at best. We know that phenylalanines in the repeats are essential because F→S or F→A mutations abolish hydrogel formation, phase separation, or indeed any cohesive interaction[29,31,43]. This suggests that hydrophobic interactions are the key. However, how such crucial phenylalanine might interact with other repeats or whether repeats interact in pairs or form higher-order clusters is still unresolved.

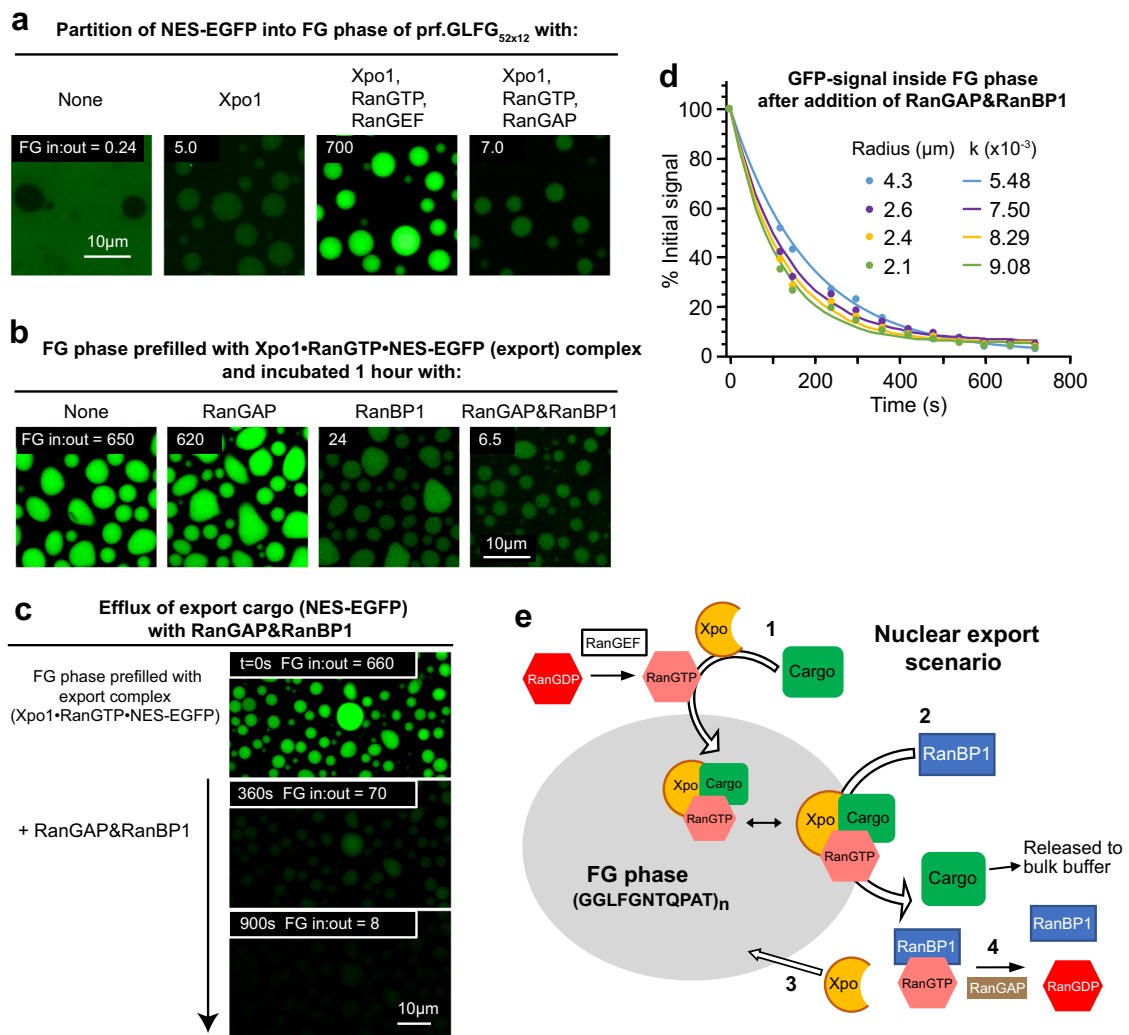

**Fig. 9 Recapitulation of RanGTPase-controlled nuclear export with the FG phase formed by the perfectly repetitive variant. a** The FG phase of prf. GLFG$_{52x12}$ was incubated for 60 min with NES-EGFP, an ATP/GTP-regenerating system, and either (i) buffer only, (ii) Xpo1, (iii) Xpo1 and wild-type RanGTP (preloaded with GTP by RanGEF), or (iv) Xpo1 and RanGDP (converted by RanGAP to the GDP form). **b** In another setup, the FG phase of prf. GLFG$_{52x12}$ was initially filled with the Xpo1·RanGTP·NES-EGFP complex in the presence of RanGEF, then incubated individually for 60 min with the indicated components, and finally imaged. **c, d** As in (**b**), the FG phase of prf.GLFG$_{52x12}$ was initially filled with the Xpo1·RanGTP·NES-EGFP complex. At time=0 s, RanGAP and RanBP1 were added to trigger NES-EGFP efflux. The signal of GFP was recorded over time and analyzed as described in Fig. 8. Source data are provided as a Source Data file. **d** A plot of GFP signals inside FG particles of different radii against time and best-fits of the single exponential decay function: $f(t) = Ae^{-kt} + B$ (see also Supplementary Fig. 9). **e** Illustration of the above processes.

The ultimate simplification of an NPC-like FG phase system, as reported in this paper, should greatly facilitate a more profound structural analysis: sequence assignments in NMR will be more straightforward, spectra easier to interpret, and peak intensities far higher if instead of 600 residues of heterogeneous repeats, a perfectly repeated 12mer GLFG sequence is analyzed. The gain of simplification will be even greater when analyzing repeat–repeat interactions. Considering just binary interactions in the wild-type sequence leaves us with thousands of possibilities to combine the >50 different repeats into pairs. With perfect repeats, there is only one.

The simplified GLFG phase would, for similar reasons, also be an ideal playground for simulations/theoretical modeling[79–81]— for gaining insights not only into cohesive interactions but also into FG interactions with mobile species that are energetically either favored or disfavored. Importantly, models for the simplified GLFG phase can be calibrated with parameters derived from this study: critical concentrations for phase separation, partition coefficients of inert species, NTRs, and NTR·cargo

complexes, as well as intra-phase mobilities of these mobile species and the FG domains themselves.

With this paper, we present one solution to the FG-simplification problem. Our starting point was the ciliate Mac-Nup98A FG domain, and we kept amino acid composition and dipeptide frequencies as close as possible to the original. Nup98 FG domains of other species can have different compositions, e.g., N/Q-rich ones in yeast or proline-rich ones in plants[32]. So, there will be multiple appropriate solutions. We assume that these alternatives still have to conform to the truly invariant and highly conserved features of Nup98 FG domains, which include a remarkably constant density of eight FG motifs per 100 residues as well as a strong selection against charged amino acids and hydrophobic residues other than Phe and Leu. Apart from this, it is well possible that we do not yet know all the crucial constraints, so exploring the space of "correct" solutions to the simplification problem will be a truly exploratory one.

On the other hand, it will be fascinating to test how the phase behavior and transport selectivity gets impacted when these

conserved features are changed. For example, it is tempting to speculate that shorter inter-FG spacers will result in smaller meshes and tighter sieving, while longer ones might relax the sieve effect. Likewise, changing the FG motifs to other sticky motifs might result in novel synthetic biomaterials with unprecedented selectivity properties. This playground might eventually also provide an answer to the evolutionary question of why an FG-based barrier serves selective nucleocytoplasmic exchange better than alternative cohesive polymers.

Nup98 FG phases function in NPCs that contain a considerable mass of less cohesive or even non-cohesive FG domains. These additional domains are typically localized at the cytoplasmic or nuclear periphery of NPCs and probably represent platforms for rapid disassembly of NTR·cargo complexes. They might also form additional filter zones to fine-tune the selectivity of NPCs. Cohesive FG-FG interactions compete with NTR-binding to FG motifs. Non-cohesive FG domains might therefore be more efficient in NTR-capture, thus functioning as "collectors" for incoming NTR-cargo complexes and increasing the capacity of NPCs for active transport. Indeed, it will be very exciting to reconstitute the cooperation of Nup98 FG domains with other FG domains and test those assumptions. A layer-wise FG phase assembly system might be the first step in this direction.

FG phases were perhaps the earliest example of reconstituted cellular condensates[29]. However, they represent just one extreme in a range of similar biological phases, which govern not only transport processes but also developmental programs, RNA-metabolism, RNP assembly, stress, and disease conditions[54,82–91]. A deeper understanding of the FG phase system is therefore likely to have broader implications.

Finally, our discovery that very established fluorescent probes commonly used in cell biology also label the FG phase of NPCs was surprising. Given that DAPI and Hoechst stain DNA exceedingly bright (Fig. 4b) and that the DNA concentration in the vicinity of NPCs is very high, the dyes' FG-philic properties are not obvious when staining cells by standard protocols. Nevertheless, we consistently observe a conspicuous signal at the nuclear periphery when treating digitonin-permeabilized cells with DAPI or Hoechst (Fig. 5), which might include a visible NPC contribution. In any case, the UV-induced effect of DAPI and Hoechst on nuclear transport is striking and opens up a wide experimental playing field.

We anticipate that such fluorescent probes will also become valuable tools for studying broader condensate biology beyond FG phases and perhaps even allow for photo-optic control of their properties and function in reconstituted systems and cells. Indeed, DAPI or Hoechst dyes are excellent lead compounds for truly FG-specific or condensate-specific reagents. Introducing substituents that sterically exclude the fluorophores from binding the DNA's minor groove could mark a possible path toward such probes.

## Methods

**Design of FG domain variant DNA sequences**. DNA sequences encoding the variants were generated with the assistance of the *Gene Designer* program, which minimized the repetition of local DNA sequences and optimized codon usage for *E. coli* expression. DNA fragments were synthesized by *GenScript* and cloned into a bacterial expression vector for overexpression and purification (see below). *Tetrahymena thermophila* GLEBS domain was kept in the amino acid sequences for most of the FG domain variants because the presence of the corresponding non-repetitive DNA sequence improved the ease of DNA syntheses.

### Recombinant protein expression and purification

*FG domain proteins*. An N-terminal histidine tag and a C-terminal Cys residue were fused to each of the FG domain sequences for protein purification and maleimide-based fluorescent labeling, respectively[32]. The proteins were expressed in a codon-optimized form in *E. coli* NEB Express at 30 °C for 3 h and induction by 0.4 mM IPTG. The expressed FG domain and variants phase-separated in vivo to form inclusion bodies, indicating their cohesiveness in a physiological environment[32].

Cells were resuspended in cold 50 mM Tris/HCl pH 7.5, 300 mM NaCl, 1 mg/ml lysozyme, and lysed by a freeze–thaw cycle followed by mild sonication. Inclusion bodies containing the FG domains were recovered by centrifugation (*k*-factor: 3158; 10 min) and washed once in 50 mM Tris/HCl pH 7.5, 300 mM NaCl, 5 mM DTT. The FG domains were extracted with 40% formamide, 50 mM Tris/HCl pH 7.5, 10 mM DTT. The extract was cleared by ultracentrifugation (*k*-factor: 135; 90 min) and applied 3 h at room temperature to a Ni(II) chelate column. The column was washed in extraction buffer, with 200 mM ammonium acetate pH 7.5. Proteins were eluted with 30% acetonitrile, 265 mM formic acid, 10 mM ammonium formate, and directly lyophilized, weighed, and dissolved to 1 mM (≈60 μg/μl) protein concentration in 4 M guanidinium hydrochloride (GuHCl). Since the absorbance of FG domains at 280 nm is very low, gravimetry is the preferred method to quantify the protein amount.

Noticeably, after the induced expression of Pro-free_prf.GLFG$_{52 \times 12}$, we found that the bacterial inclusion bodies containing the FG domain stain bright with Thioflavin-T (ThT), indicating an intrinsic amyloid-forming propensity of the Pro-free_prf.GLFG$_{52 \times 12}$. The FG domain in the inclusion bodies was insoluble in 40% formamide or 4 M GuHCl and had to be extracted with 6 M GuHCl, 50 mM Tris/HCl pH 8.0, 10 mM DTT.

The Ni-eluate containing the *sc*Nup116 FG domain was applied to a Tosoh TMS C1 column (10 mm). The column was washed in Buffer A (0.08% TFA, 10% acetonitrile) and eluted in a gradient ending at 30% acetonitrile. Fractions containing the full-length protein were pooled, lyophilized, and finally dissolved to 1 mM protein concentration in 4 M GuHCl[6].

*Protein probes for FG particle permeation assays*. Most NTRs and other proteins (including GFP variants, mCherry and RanGTP/GDP) were expressed as His-tagged-fusions (Supplementary Table 4) and purified by native Ni(II) chelate chromatography, as described previously[6,32]. Elution was performed with either imidazole or by on-column SUMO protease cleavage[92]. For the former, the tags of the imidazole-eluted proteins were cleaved off in solution with TEV protease, and proteins were further purified by gel filtration on a Superdex200 column equilibrated with 44 mM Tris pH 7.5, 290 mM NaCl, 4.4 mM MgCl$_2$, 5 mM DTT, and eventually snap-frozen in liquid nitrogen after addition of 250 mM sucrose.

(See Supplementary Table 4 for a list of proteins and the corresponding bacterial expression constructs used in this study.)

**Fluorophores**. Abberior STAR 635P maleimide was purchased from Abberior (Germany). Alexa Fluor 488 C5 maleimide, Alexa Fluor 568 C5 maleimide, Alexa Fluor 594 C5 maleimide, Alexa Fluor 647 C2 maleimide, DyLight405 maleimide, DyLight488 maleimide, Oregon Green 488 maleimide, and SYPRO orange were purchased from ThermoFisher Scientific (Germany). Atto488, Atto647N, and Atto565 maleimides were purchased from ATTO-TEC (Germany). Cy3, Sulfo-Cy3, Cy5, and Sulfo-Cy5 maleimides were purchased from Lumiprobe (Germany). All maleimides were incubated with a 10× excess molar ratio of L-cysteine for 1 h for quenching before the FG phase permeation assays. DAPI, Hoechst 33342, Hoechst 33258, Hoechst 34580, Oxazole yellow, Ethidium bromide, Fluorescein, Rhodamine B, and Thioflavin-T (ThT) were purchased from Sigma-Aldrich (Germany). Nuclear yellow was purchased from Abcam (Netherlands).

**FG phase (particle) preparation for permeation assays**. A previously described procedure[32] was applied with minor modifications. Typically, 2 μl of a 1 mM (≈60 μg/μl) FG domain solution (in 4 M GuHCl) was rapidly diluted with 100 μl assay buffer (50 mM Tris/HCl pH 7.5, 150 mM NaCl, 5 mM DTT) and 7.5 μl of the suspension was mixed with 22.5 μl substrate (typically 1 μM NTRs or 6 μM mCherry in assay buffer, see also below). The resulting mixture ([FG domain] = 5 μM) was placed on collagen-coated μ-slides 18-well (IBIDI, Germany) that had been further passivated with 0.1 mg/mL MBP. Before imaging, FG particles were allowed to sediment for 60 min to the bottom of the slide.

**Confocal laser-scanning microscopy**. mCherry and GFP/Alexa488 signals were acquired with 561 and 488 nm excitation, respectively, using a Leica SP5 or SP8 confocal scanning microscope equipped with a ×63 oil immersion objective and hybrid detectors (standard mode, in which nonlinear response of the detector was auto-corrected) at 21 °C. For covering wide dynamic ranges, scan settings (e.g., laser power) for FG particle permeation assays were adjusted individually, and quantification values under non-saturating conditions are shown in the figures.

The following probe concentrations were used in FG particle permeation assays: mCherry at 5 μM; protein probes with partition coefficients <2 (efGFP8Q and sffrGFP4 25xR→K) at 2 μM; protein probes with partition coefficients ≥2 at 0.75 μM, Thioflavin-T at 2 μM, DAPI at 9 μM, Hoechst derivatives at 9 μM; EtBr at 2.5 μM and other fluorophores at 2 μM.

Thioflavin-T was excited at 405 nm and detected in a 460–500-nm window. DAPI/Hoechst derivatives were excited at 405 nm and detected in a 410–550-nm window. EtBr was excited at 488 nm and detected in a 590–642-nm window. Other fluorophores were detected according to the instructions of the suppliers. Detection windows for the three-channel imaging described in Fig. 4f were: Hoechst/410–470 nm, GFP/500–555 nm, and mCherry/577–700 nm. Channels were scanned sequentially to minimize cross-channel spillover of signals.

**Quantification of partition coefficients/FG in:out ratios**. For each imaged field, raw signals within 3–5 FG particles (IN) and three reference areas within the buffer (OUT) were quantified using Leica LAS-AF software. For cases where the fluorophore is insensitive to the environment (e.g., Alexa488/GFPs/mCherry), partition coefficients (Part. Coeff.) into each particle were calculated according to:

$$Part.Coeff. = IN/Mean(OUT)$$

As reported previously[6], standard deviations were typically <10% between individual particles and between experiments. Mean values are shown in the figures.

Note that for environmentally sensitive fluorophores (e.g., ThT/Hoechst/DAPI/SYPRO orange), the fluorescence ratio in:out FG phase does not reflect the partition coefficient of a given fluorophore directly because the fluorescence was enhanced inside the FG phase. The FG in:out ratios were indicated instead.

**Bulk fluorescence measurements**. FG phases corresponding to 400 µg of FG domain protein were prepared in 150 µl 50 mM Tris/HCl pH 7.5, 150 mM NaCl, as described above. Then 20 µM of fluorophore (Hoechst 33342/Hoechst 34580) was added. Reads for Hoechst signals were recorded after 30 min. Fluorescence signals in Fig. 4b were detected in a wavelength 440–480-nm window; excitation was at 360 nm (by Synergy H4 microplate reader, BioTek; wavelengths were selected by bandpass filters). "Plasmid" in Fig. 4 was pUC57 (2710 bp). Error bars were standard deviations of three replicates. Emission spectra were recorded by monochromator-based measurements at excitation wavelengths specified in Fig. 4 at a step size of 5 nm. The instrumental background was recorded by measuring just assay buffer and subtracted from all raw data.

ThT fluorescence measurement in Supplementary Fig. 1: FG phases corresponding to 300 µg of FG domain proteins were prepared in the buffer described above. In total, 20 µM Thioflavin-T (ThT) was added 2 min after phase-separation and fluorescent signals were recorded at 2-min intervals (excitation: 446 nm; detection: 477–487 nm).

**NPC-blocking tests with UV exposure and DAPI/Hoechst**. HeLa-K cells were grown in Dulbecco's Modified Eagle's Medium (DMEM, high-glucose), supplemented with fetal calf serum (FCS) and antibiotics ("AAS", Sigma-Aldrich) on eight-well µ-Slides (IBIDI, Germany) to 70–90% confluency. Plasma membranes were permeabilized by treating the cells with 20 µg/ml digitonin in transport buffer (20 mM HEPES/KOH pH 7.5, 110 mM potassium acetate, 5 mM magnesium acetate, 0.5 mM EGTA, 250 mM sucrose) for 3 min at 23 °C (with gentle shaking), followed by three washing steps in transport buffer. Then, 5 µM of either DAPI or Hoechst 33342 in transport buffer was added for DNA staining. A Leica SP8 confocal laser-scanning microscope was used for imaging. Where indicated, a region of a well was exposed to UV light at the lowest possible intensity setting for 30 s, while other regions remained non-exposed. As a UV source, an EL6000 mercury metal halide lamp (Leica, Germany) equipped with a bandpass 360/40-nm excitation filter was used. Next, before imaging, the DNA-staining solution was exchanged for the nuclear import mix (2 µM mCherry, 1 µM IBB-sinGFP4a, 0.5 µM human importin β) in transport buffer supplemented with components of the Ran system (4 µM Ran, 0.8 µM NTF2, 0.4 µM RanGAP, 0.05 µM RanBP1) and an ATP/GTP-regenerating system (0.5 mM ATP/GTP, 10 mM neutralized phosphoenolpyruvic acid, 0.5 µM human pyruvate kinase, 5 µM E. coli nucleoside-diphosphate kinase). Import was followed real-time and the ≈200-s time point is shown in Fig. 5. A population of digitonin-permeabilized cells always contained a certain fraction of hyper-permeable nuclei (≈5–20%). These were identified by the criterion of their permeability toward mCherry and excluded from the analysis.

**Analyses of phase separation**. In total, 1 µl of a 1 mM FG domain solution (in 4 M guanidinium-HCl) was rapidly diluted with 100 µl assay buffer (50 mM Tris/HCl pH 7.5, 150 mM NaCl, 5 mM DTT), i.e., [FG domain]=10 µM. After 1 min, the FG phase (insoluble content) was pelleted by centrifugation (21130 g, 30 mins, 25 °C (using a temperature-controlled Eppendorf 5424R centrifuge equipped with a FA-45-24-11 rotor).

Equivalent ratios (6%) of pellets (FG phases) and supernatants were analyzed by SDS-PAGE/Coomassie blue staining. Critical concentration for phase separation of a given FG domain was taken as the concentration that remained in the supernatant, which was estimated with a concentration series loaded onto the same gel.

**Immobilization of FG particles and washing**. In total, 2 µl of 1 mM FG domain solution (in 4 M guanidinium-HCl) premixed with 2.5% Alexa647-coupled FG domain molecules was diluted 50-fold by assay buffer to allow phase-separation, then 75 µl (≈100 µg) of the FG particles was overlaid on cushion buffer (300 µl assay buffer with 30% sucrose) on a glass coverslip. Mild centrifugation (150 g, 5 min) immobilized the FG particles on the coverslip, which was then assembled into a flow chamber. The FG particles were then washed once with fresh assay buffer and preloaded with 2 µM Alexa488-labeled NTF2. Continuous washing cycles were performed using the flow chamber (1 cycle≈400 µl fresh buffer for 30 s). Signals of Alexa488 and Alexa647 after each cycle were recorded by confocal scans.

**Fluorescence recovery after photobleaching (FRAP) and estimation of diffusion coefficients**. FG particles ($n = 5$ typically) preloaded with indicated protein probes were photobleached at 488 nm at 21 °C. After bleaching, GFP/Alexa488 signals were acquired at 2 s of intervals. Raw data were corrected for photobleaching during acquisition and normalized to the initial signal. Bleached areas were manually defined to a circular region inside the FG phase with a diameter of 3 µm; however, we found that the actual bleached area was always ≈4–7 µm in diameter, due to the diffusion during bleaching and Gaussian blurring of the laser beam[93]. The diffusion coefficients ($D$) were estimated with (i) the measured bleached radii ($r$, taken as half of the bleached width at 86% of the bleached depth) and (ii) half-times ($t_{1/2}$) of the recovery of the bleached area (the datasets were fitted to exponential growth curves using SigmaPlot using the non-linear least-squares fitting), by the analytical approach and expressions described before using a 2D diffusion model[94]:

$$D = 0.224 \bullet r^2/t_{1/2} \qquad (1)$$

**Estimation of diffusive times and distances from diffusion coefficients**. Diffusion coefficients obtained in this study were correlated to the time required to traverse a distance (root mean square displacement, r.m.s.d.) of 2 nm or 10 nm (see the main text for the rationale); and distance traveled within 10 milliseconds (Supplementary Table 3): the typical timescale of a nuclear translocation event, by Fick's laws of diffusion and a 1D random walk model:

$$\bar{d} = \sqrt{2Dt} \qquad (2)$$

or written as: $t = \frac{(\bar{d})^2}{2D}$ where $\bar{d}$ is the root mean square displacement from the starting point. $D$ is the measured diffusion coefficient and $t$ is the time.

**Efflux of import cargo from FG phase**. In total, 2 µl of 1 mM FG domain solution (in 4 M Guanidinium-HCl) was rapidly diluted 50-fold by assay buffer to allow phase-separation. In all, 10 µl of the suspension was mixed with 5 µl of the substrate (containing 4.5 µM hsImpβ and 3 µM hsIBB-EGFP in assay buffer). The resulting mixture was placed on MBP/collagen-coated µ-slides, and FG particles were allowed to sediment under gravity for 60 min (and for reaching an equilibrium state). In all, 15 µl of 14 µM MBP-mCherry-RanQ69L in assay buffer (i.e., concentration after mixing = 7 µM, see also Supplementary Fig. 7) was added to trigger the efflux of IBB-EGFP from FG particles and the signals of GFP/mCherry were recorded over time, as described above. Datasets in Fig. 8c, d were obtained from a single image set. All datasets were fitted to single exponential decay curves by a GRG nonlinear solving method in Solver, Microsoft Excel.

**Influx and efflux of export cargo**. In total, 2 µl of 1 mM FG domain solution (in 4 M guanidinium-HCl) was diluted 50-fold by "Mg²⁺ assay buffer" (50 mM Tris/HCl pH 7.5, 150 mM NaCl, 5 mM DTT, 2 mM MgCl₂) to allow phase-separation. In all, 15 µl of the suspension was mixed with 15 µl of "Substrate A" (containing 10 µM RanGTP/GDP, 1 µM RanGEF or RanGAP, 1 µM NES(PKI)-EGFP, 2 µM hsXpo1, and an ATP/GTP-regenerating system, in Mg²⁺ assay buffer; or see specifications in Fig. 9). The resulting mixture was placed on MBP/collagen-coated µ-slides, and FG particles were allowed to sediment under gravity for 60 min and then imaged. In Fig. 9c, d, the supernatant was exchanged for 30 µl of "Substrate B" (containing 4 µM RanBP1, 4 µM RanGAP (or as specified), and 2 mM GDP in Mg²⁺ assay buffer) to trigger the efflux of NES-EGFP from FG particles. The signal of GFP was subsequently recorded over time, as described above. Datasets in Fig. 9d were obtained from a single image set.

**Reporting summary**. Further information on research design is available in the Nature Research Reporting Summary linked to this article.

## Data availability

The data that support the findings of this study are available from the corresponding author upon reasonable request. Source data are provided with this article.

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

## Acknowledgements

We thank Broder Schmidt and Steffen Frey for introducing S.C.N. to the FG phase topic and sharing permeation probes, Jürgen Schünemann for reverse-phase purification of FG domains, Łukasz Jaremko for support with evaluating FRAP data and ThT fluorescence kinetics and helpful discussions, former lab members for sharing plasmids, Connie Paz for proofreading as well as the Max-Planck-Gesellschaft and the Deutsche Forschungsgemeinschaft (SFB 860) for funding.

## Author contributions

S.C.N. designed and performed most of the experiments. T.G. designed and performed the UV-induced blocking experiments of NPCs, and prepared the components of the nuclear export complex. D.G. conceived the overall concepts of the study, designed the FG domain variants, and wrote the paper. All authors contributed to data analysis, interpretation, and manuscript writing.

## Funding

## Competing interests

The authors declare no competing interests.
