## [Peer Review File · Nature Communications]

Reviewers' Comments:

Reviewer #1:

Remarks to the Author:

In this manuscript, Ng et al. aim at engineering an artificial barrier in vitro to mimic what happened in the native nuclear pore by using extracted GLFG12mer peptides and then testing the selectivity of the GLFG-assembled barrier in nuclear transport. In more detail, there are three major steps included in the draft: 1) in vitro mimic of the NPC's selectivity barrier by using GLFG domains; 2) test importin- and exportin-mediated cargo transport through this artificial barrier; and 3) use DAPI/ Hoechst dyes to block nuclear transport after UV-photo-induction. The major contribution of this study is mainly from the first two measurements, however, the similar experiments and conclusions have been published from the corresponding author's lab 15 years ago (Frey et al. *Science*, 314:815, 2006). Also, the manuscript was poorly written by having many inaccurate and confusing statements, which made it hard to read. Although substantial data was presented, more importantly, both the significance and innovations of this manuscript are very limited. Thus, this reviewer doesn't support to publish it in *Nature Communications*.

The manuscript was not well organized by including numerous errors. Only some of the major and minor issues are listed below:

1) Given its relatively small size, NTF2 (30 kDa) is likely to freely diffuse into the Nucleus when not bound to RanGAP. This makes the assumption that this transport factors interaction with the FG motifs is evidence of a selectively permeable FG region imperfect. Given that the key thesis of this paper is to reproduce the selectivity barrier, it is problematic that they have chosen a nuclear transport factor smaller than the diffusion barrier.

2) In the background, the author references irregular inter-FG spacers. Then in the results section, design their experimental FG region to move hydrophobic residues from within the inter-FG spacers to the FG motifs. The paper would benefit from a brief explanation in the background explaining why (if at all) FG spacers are important. Authors state they adjusted inter-FG spacers to be an identical length. What length? This is important, as they moved the hydrophobic amino acids from the spacers to the FG regions, and the distance between the regions will provide insight into the efficacy of the system being a result of the FG/FG-like motifs or if it is a byproduct of a concentration of hydrophobic amino acids. Are the spacer regions intrinsically disordered?

3) As for the experiments on NTF2, there are many more issues to be taken care of as follows:

A) It should be specified in the text that the NTF2 is tagged with Alexa488;

B) The FG regions are derived from algae (*Tetrahymena thermophila*) and the NTF2 is derived from Rat. It is unclear how similar the engineered FG regions are to the native FG regions within Rats.

C) Where on the NTF2 is the Alexa488 binding? Are they using an antibody labeling strategy or is the NTF2 covalently bound to the Alexa488?

D) This is critical to know as fig 1 and 2 depict key evidence for their central thesis, specifically, that their FG aggregates selectively interact with NTF2, but not mCherry. Depending upon where the Alexa488 is associated with NTF2, the FG interaction region could be occluded. This would then imply that the interaction is driven by Alexa488 (a highly charged and prolific binder) and not NTF2.

E) Transport times are compared to Imp-B1 import, are dynamics of NTF2 dynamics known in any capacity? The comparison of import dynamics needs to recognize how imperfect of a comparison this is.

4) This sentence "The selective phase of NPCs is just one extreme in a range of similar biological condensates that govern developmental programs, RNA-metabolism, RNP assembly, stress, and disease conditions^{30–37}." was listed as a separated paragraph? Is this a typo or real?

5) Many statements are inaccurate and confusing throughout the manuscript. For example, in the

abstract it says that "NPCs contain numerous distinct FG domains, each comprising variable repeats.", but soon after in the section of introduction, the authors stated that "Vertebrate NPCs contain ≈ 10 different FG domains". It's hard to guess which one that the authors want readers to follow.

6) It might be too arbitrary and inaccurate by stating that "the barrier can be described as a condensed phase assembled from cohesive FG repeat domains." because there are still disputes about the assembly of the NPC's selectivity barrier and also many experiments argued against the hypothesis of "a condensed phase" in the NPC.

7) It is stated that Serine is used to replace Proline in the proline free FG sequence. However, the logic for why they chose to remove proline is not readily apparent to me.

8) The results highlight that the contents of the inter-FG spacers do have an impact upon function of the FG region. The background section should include a statement summing up current thinking on the function of the inter-FG spacers.

9) Authors state that the hydrophobic side chain on the surface of GFP enhances FG interactions, this is to be expected. That said, why is there an explanation why the relatively hydrophobic mCherry does not interact with the FG phases? (source for mCherry being much more hydrophobic than DsRed: <https://www.ncbi.nlm.nih.gov/pmc/articles/PMC3394910/>)

Reviewer #2:

Remarks to the Author:

Ng et al create a synthetic nuclear transport permeability barrier with minimalistic composition. In four intermediates steps they created a synthetic protein that contains 52 identical GLFG_12mers, all of which phase separate in a similar way as compared to wt protein. Sequence variation affected amyloid forming propensity. The authors further identify a contribution of the GLEBS domain to phase separation. Various experiments proof that the engineered FG-phase recapitulates the cargo/NTR entry and release behavior of 'non-engineered' FG-Nups.

A very laudable innovation is the demonstration of cargo complex release upon addition of RanGTP-RFP. Also the Ran-GTP facilitated entry of an export complex is a very nice experiment! Those data finally show that in vitro FG-phases also recapitulate the most essential steps of active nuclear transport.

Otherwise, different fluorescent dyes are tested for entry into the phase. The authors demonstrate that dye attachment to proteins entering the permeability barrier affects their partitioning coefficient and that NPCs can be inactivated by illumination, both of which are very neat experiments and important for the field.

The paper is very interesting. Why there are so many different types of FG repeats is a major outstanding question. The demonstration that this variation, although conserved, is not required to recapitulate active nuclear transport in vitro, comprises an important step forward. As always, the paper is technically well done and taken together, a strong candidate for publication in Nat Comm after some revisions.

Some comments:

Having that said, the weakness of the paper is that it is solely based on an in vitro approach. Engineering an in vivo NPC that would only contain a single engineered FG-Nup is challenging. Although seemingly feasible in yeast if done gradually, it would be lots of work and thus beyond the scope of the present study. However, one really wishes to know if a yeast strain with all the same FG

repeats would be viable or alternatively, gradually develop fitness defects once native FG-Nups are changed one by one. You could at least refer to Strawn et al, NCB 2004 in the discussion. I understand that this is not the exact same idea. It is still interesting in this context that many repeats can be removed in vivo and cells are still viable.

Style: As pointed out above, this reviewer appreciates the middle part of the manuscript, but it does interrupt the flow of the paper. I do not have a good suggestion where to place it elsewhere but you may consider trimming it down – or integrate it better.

Page 13: Why did the authors use an RFP fusion of Ran? This should perfectly work with native RanQ69LDeltaC-GTP, correct? It would be important to know if release becomes faster, consistent with the idea that non-tagged Ran can diffuse into the phase – or not.

Introduction on page 4: While the 3rd paragraph emphasizes the remarkable sequence conservation of Nup98FG, the 4th paragraph states that ‘substantial progress has been hampered by the sequence diversity within given Nup98FG domains’. For readability, you may want to include an additional sentence explaining the concept of comparing sequences of FG repeats within the same protein to contrast previous paragraph better (most people think in terms of comparing sequences across proteins).

Related to that, if the different properties of individual repeats are conserved across different species but a simple repeat recapitulates nuclear transport, what could be the selection pressure?

Some of the authors have previously shown that MacNup98 is bit special in the sense that it contains many glycines and forms a very tight hydrogel already at comparably low intra-particle FG domain concentrations (eLife paper from 2013). Why is it a well-justified choice for this study that hopefully allows for generic conclusions? The statement that it is well-characterized is understandable. However, it makes the reader wonder about the relevance, because the source species is not exactly a main stream model organism. A bit of additional introduction may be helpful.

The authors state that they expect Hoechst dyes to become useful for staining FG-phases in vitro, but they also show that even a single dye attached to GFP has a quite massive effect. This does not come across very sound and maybe explained better. What type of experiments could be done despite the fact that the dye massively changes the given phase?

This reviewer does not like the term ‘authentic NPC’, because it inversely suggests that phase separated FG-bodies are ‘non-authentic NPCs’. Why can’t it just be an ‘NPC’?

Reviewer #3:

Remarks to the Author:

The underlying mechanisms governing the selective barrier imposed by the nuclear pore complex (NPC) remains to be fully understood. This paper takes an innovative protein-engineering approach to define a “simple” sequence that can recapitulate the fundamental selective sorting properties of the NPC in vitro. By beginning with a Nup98 FG-domain from Tetrahymena, the authors sequentially minimize the sequence to a near perfect repeating GLFG peptide. A critical control is a similar peptide lacking proline residues. Using these reagents, they demonstrate that they can form cohesive networks in vitro that phase-separate into condensates that can recapitulate both active import and export while excluding an inert mCherry protein. Along the way, they also interrogate how these condensates interact with a series of fluorescent probes. The major advance here is that they demonstrate that DAPI can crosslink the FG-network, which could provide a useful tool for the field. In general, the data are of high quality and are rigorous. The manuscript is written well although it is a bit disjointed due to the “detour” exploring interactions with probes. It should make a valuable

contribution to Nature Communications as it will provide valuable concepts and tools for multiple fields.

Major Point 1: The paper is written with considerable tunnel vision as to how the NPC acts as a selective barrier as it is solely focused on the importance of cohesive interactions. It would be nice if the authors at least considered that there are non-cohesive FG-networks in the NPC that are known to be important functionally. No additional data is required, but a discussion of the disconnect, for example, between the ability of the GLFG-domain to exclude mCherry in vitro and the observation that mCherry can freely pass through the NPC in vivo would seem warranted, particularly in the context of work defining the NPC barrier in vivo as being "soft" (e.g. Timney, JCB).

Major Point 2: Hydrogel versus liquid? A description of the condensates in terms of their physical characteristics would be helpful. Although it is acknowledged that there is a continuum of physical states between liquids and gels, there is also the concept that cohesive interactions can transition or "age" between liquids and more solid states, the latter being less functional/pathological. The photobleaching in Fig. 6D may, for example, be indicating that the GLEBS domain drives a gel-like state. Again, I don't think any additional experimentation is needed, but some discussion of this in the context of the phase-separation literature should be considered.

Major changes to the manuscript:

- We have expanded the first part of the introduction to make clear that NPCs display great transport selectivity already towards GFP-sized mobile species. This addresses comments by reviewers 1 and 3.
- We have expanded the Introduction to cover also non-cohesive FG domains. This addresses comments by reviewers 1, 2, and 3.
- We have included three additional \pm NTR controls (Figure 6F-H) to document the transport selectivity of our perfectly repeated FG phase. The effects are quite striking, covering factors of 200 to 3500. This addresses a concern of reviewer 1.
- We expanded the Perspectives by a paragraph on non-cohesive FG domains and their possible cooperation with the Nup98 phase system.

Answers to the reviewers' comments

(for clarity, we repeat their points in blue in front of each of our replies)

Reviewer #1 (Remarks to the Author):

In this manuscript, Ng et al. aim at engineering an artificial barrier in vitro to mimic what happened in the native nuclear pore by using extracted GLFG12mer peptides and then testing the selectivity of the GLFG-assembled barrier in nuclear transport. In more detail, three are three major steps included in the draft: 1) in vitro mimic of the NPC's selectivity barrier by using GLFG domains; 2) test importin- and exportin-mediated cargo transport through this artificial barrier; and 3) use DAPI/ Hoechst dyes to block nuclear transport after UV-photo-induction. The major contribution of this study is mainly from the first two measurements, however, the similar experiments and conclusions have been published from the corresponding author's lab 15 years ago (Frey et al. Science, 314:815, 2006).

This is an incorrect account of our 2006 Science paper, where we demonstrated for the first time cohesive FG domain interactions, FG hydrogel formation and provided evidence for inter FG cohesion being required for viability. However, the paper did not contain a single piece of data addressing transport selectivity and did not contain any experiment to simplify the sequence of the FG domain while keeping it cohesive, nor any attempt to show that sequence heterogeneity of the native FG domain is not important.

Also, the manuscript was poorly written by having many inaccurate and confusing statements, which made it hard to read. Although substantial data was presented, more importantly, both the significance and innovations of this manuscript are very limited. Thus, this reviewer doesn't support to publish it in Nature Communications.

The manuscript was not well organized by including numerous errors. Only some of the major and minor issues are listed below:

1) Given its relatively small size, NTF2 (30 kDa) is likely to freely diffuse into the Nucleus when not bound to RanGAP. This makes the assumption that this transport factors interaction with the FG motifs is evidence of a selectively permeable FG region imperfect. Given that the key thesis of this paper is to reproduce the selectivity barrier, it is problematic that they have chosen a nuclear transport factor smaller than the diffusion barrier.

First, there is no reason to believe that RanGAP binds NTF2. RanGAP triggers RanGTP to hydrolyze its nucleotide, while NTF2 mediates the nuclear import of RanGDP.

Second, it is an error to assume that 30kDa proteins *freely* diffuse through NPCs. Instead, nuclear pores are highly selective already at this size level (see: Frey et al., 2018 and references therein). We have now expanded the first part of the introduction to explain this point.

NTF2 (29 kDa) passes NPCs ~700 times faster than mCherry (26.5kDa) and even 6000 times faster than the most surface-inert GFP variant (26.7kDa). These differences in NPC-passage rates are remarkably well represented by differences in FG phase-partition coefficients.

There is a straightforward explanation of why NPCs need to be selective towards 20-30kDa-sized proteins, which relates to Ran (25kDa). NPCs need to suppress the dissipation of the nucleocytoplasmic RanGTP-gradient, thus restricting the efflux of RanGTP and ensure that only NTR-bound RanGTP exits cell nuclei (otherwise, the coupling between GTP-hydrolysis with transport would break down). This concept is supported by the fact that RanGDP subsequently employs NTF2 for efficient retrieval into nuclei. Thus, the comparison of 30kDa mobile species is highly relevant for NPC function and was consciously chosen.

The reviewer is also incorrect in implying that we had restricted our analysis to small mobile species. He/ she missed that we also studied the tetrameric GFP^{NTR}_3B7C (110ka), Importin β -IBB GFP complexes (130kDa), Xpo1-export complexes (170kDa), as well as importin β -IBB-GFP-MBP-M9-transportin complexes (270kDa) (in Figs. 6, 8 and 9).

2) In the background, the author references irregular inter-FG spacers. Then in the results section, design their experimental FG region to move hydrophobic residues from within the inter-FG spacers to the FG motifs. The paper would benefit from a brief explanation in the background explaining why (if at all) FG spacers are important.

The spacers are foremost required for adjusting the overall FG-density within the FG domains. Spacer-less (GLFG)_n polymers will form a very compact phase that is impossible to cross by any NTR.

Different FG domains feature different types of spacers, which in turn can have a profound impact on the phase behavior. Highly charged spacers, for example, confer high water solubility and counteract cohesive interactions. We have expanded on that in the Introduction, Results and Discussion, as well as in the Perspectives.

Authors state they adjusted inter-FG spacers to be an identical length. What length?

Repeat units have been adjusted to a length of 12 residues. If we define “GLFG” as the FG motif, this leaves eight residues for the spacers. A narrower definition of “FG” as the FG motif leaves ten residues as a spacer. This should be clear from Fig.1A (apparently missed by this reviewer) and the text.

This is important, as they moved the hydrophobic amino acids from the spacers to the FG regions, and the distance between the regions will provide insight into the efficacy of the system being a result of the FG/FG-like motifs or if it is a byproduct of a concentration of hydrophobic amino acids. Are the spacer regions intrinsically disordered?

The entire domain is intrinsically disordered, and so are the spacers.

3) As for the experiments on NTF2, there are many more issues to be taken care of as follows:

A) It should be specified in the text that the NTF2 is tagged with Alexa488;

The fact that NTF2 has been labelled has been mentioned four times in the figure legends as well as in the Methods and the supplements.

B) The FG regions are derived from algae (*Tetrahymena thermophila*) and the NTF2 is derived from Rat. It is unclear how similar the engineered FG regions are to the native FG regions within Rats.

Tetrahymena is a ciliate and, as such, evolutionary very distant from algae. Rat FG domains are no subject of this study, one reason being that their analysis is complicated by O-GlcNAc modifications that modulate inter-FG cohesion.

We previously surveyed Nup98 FG phases from a very wide range of eukaryotic species (mammals, lancelets, insects, nematodes, fungi, plants, amoebas, ciliates, and excavates) and found not only rather similar phase properties but also rather similar partition coefficients for both yeast and mammalian NTF2. A species mismatch between NTR and FG phase is therefore unlikely to be a serious issue.

C) Where on the NTF2 is the Alexa488 binding? Are they using an antibody labeling strategy or is the NTF2 covalently bound to the Alexa488?

We have used maleimide chemistry for labeling as previously described (Ribbeck & Görlich, 2001) and with a labeling density of 0.8 dye molecules per NTF2 dimer. The dimer contains six cysteines, and we assume random labeling of those.

An antibody labeling would not have been practical as 150kDa-sized immunoglobulins cannot enter the FG phase.

D) This is critical to know as fig 1 and 2 depict key evidence for their central thesis, specifically, that their FG aggregates selectively interact with NTF2, but not mCherry. Depending upon where the Alexa488 is associated with NTF2, the FG interaction region could be occluded. This would then imply that the interaction is driven by Alexa488 (a highly charged and prolific binder) and not NTF2.

This is highly unlikely for several reasons:

First, the effect of a single Alexa488 moiety is far too weak to explain the NTF2 partition coefficient of 2000. When labeling efGFP_{8Q}-Cys with Alexa488 maleimide, we see no more than a 3-fold increase in partition coefficient (see Figure to the right).

The NTF2 structure is consistent with the assumption that all six cysteines are part of FG-binding sites so that the modification might cause some loss as well as some gain in FG-philicity, the result being perhaps a rather neutral effect.

Second, the new figure 6F demonstrates that also unlabelled NTF2 interacts strongly with the Perf.GLFG_{12mer} phase and increases the RanGDP partition coefficient 170-fold.

Third, we did not rely blindly just on labeling with unshielded fluorophores. Instead, we also used GFPs as probes, whose fluorophores are fully buried within the beta-barrel of the protein. This includes surface-engineered GFPs that behave like NTRs (compare e.g. mCherry with GFP^{NTR}-3B7C in Figure 6B) as well as NES and IBB-GFP fusions that show a striking NTR-mediated influx into the FG phase (Figs.6B,6G, 8, and 9).

Please also note that negative charges actually *counteract* a partitioning into a Nup98 FG phase. Please see Frey et al., 2018 and Discussion therein. See also Supp.Fig. S3, which compares Cy3 and Cy5 with their more negatively charged Sulfo-Cy3 and Sulfo-Cy5 variants.

E) Transport times are compared to Imp-B1 import, are dynamics of NTF2 dynamics known in any capacity?

Dwell times of importin β -cargo complexes (~10 msec) and NTF2 (~4 msec) at NPCs have been reported previously by other groups (Yang et al. 2005; Kubitscheck et al., 2005; Yang and Musser, 2006).

The comparison of import dynamics needs to recognize how imperfect of a comparison this is.

It is unclear to us which comparison and what imperfection the reviewer is referring to.

4) This sentence “The selective phase of NPCs is just one extreme in a range of similar biological condensates that govern developmental programs, RNA-metabolism, RNP assembly, stress, and disease conditions^{30–37}.” was listed as a separated paragraph? Is this a typo or real?

This is “real”, but is has now been moved to the Perspectives section.

5) Many statements are inaccurate and confusing throughout the manuscript. For example, in the abstract it says that “NPCs contain numerous distinct FG domains, each comprising variable repeats.”, but soon after in the section of introduction, the authors stated that “Vertebrate NPCs contain ≈ 10 different FG domains”. It’s hard to guess which one that the authors want readers to follow.

“Numerous” was perhaps not the most appropriate term. We changed it to “several”. Thank you for pointing out this inaccurate phrase. It is unclear why this reviewer used the term “many” but identified just one.

6) It might be too arbitrary and inaccurate by stating that “the barrier can be described as a condensed phase assembled from cohesive FG repeat domains.” because there are still disputes about the assembly of the NPC’s selectivity barrier and also many experiments argued against the hypothesis of “a condensed phase” in the NPC.

It would have helped to detail those many experiments that argue against the hypothesis of a condensed FG phase forming the NPC permeability barrier. This would have allowed for a fact-based discussion. We wish to add that the barrier is defined by function and that this definition by no means implies that all FG domains are necessarily assembled as a condensed phase. There are also non-cohesive domains or sub-domains.

7) It is stated that Serine is used to replace Proline in the proline free FG sequence. However, the logic for why they chose to remove proline is not readily apparent to me.

The logic stems from matching amino acid compositions in the perfectly repeated domains as closely as possible to the starting wildtype sequence. The closest match is the Perf.GLFG12mer, which, however, overrepresents proline (one occurrence instead of 0.58 per repeat). The next best match to the authentic composition was, therefore, to replace proline with another residue. This rationale has been described in the text, and after re-reading this, we still feel that this was clearly presented.

8) The results highlight that the contents of the inter-FG spacers do have an impact upon function of the FG region. The background section should include a statement summing up current thinking on the function of the inter-FG spacers.

We added a summary about the inter-FG spacers to the introduction as well as two paragraphs to the Results and Discussion and the Perspectives.

9) Authors state that the hydrophobic side chain on the surface of GFP enhances FG interactions, this is to be expected. That said, why is there an explanation why the relatively hydrophobic mCherry does not interact with the FG phases? (source for mCherry being much more hydrophobic than DsRed: <https://www.ncbi.nlm.nih.gov/pmc/articles/PMC3394910/>)

Only solvent-accessible residues should matter for FG phase entry/ exclusion. The quoted paper studied something different, namely the environments of the fully buried fluorophore of the two red fluorescent protein variants. Surface-wise, mCherry is very hydrophilic.

Reviewer #2 (Remarks to the Author):

Ng et al create a synthetic nuclear transport permeability barrier with minimalistic composition. In four intermediates steps they created a synthetic protein that contains 52 identical GLFG 12mers, all of which phase separate in a similar way as compared to wt protein. Sequence variation affected amyloid forming propensity. The authors further identify a contribution of the GLEBS domain to phase separation. Various experiments proof that the engineered FG-phase recapitulates the cargo/NTR entry and release behavior of 'non-engineered' FG-Nups.

A very laudable innovation is the demonstration of cargo complex release upon addition of RanGTP-RFP. Also the Ran-GTP facilitated entry of an export complex is a very nice experiment! Those data finally show that in vitro FG-phases also recapitulate the most essential steps of active nuclear transport.

Otherwise, different fluorescent dyes are tested for entry into the phase. The authors demonstrate that dye attachment to proteins entering the permeability barrier affects their partitioning coefficient and that NPCs can be inactivated by illumination, both of which are very neat experiments and important for the field.

The paper is very interesting. Why there are so many different types of FG repeats is a major outstanding question. The demonstration that this variation, although conserved, is not required to re-capitulate active nuclear transport in vitro, comprises an important step forward. As always, the paper is technically well done and taken together, a strong candidate for publication in Nat Comm after some revisions.

Thank you very much for this enthusiastic summary!

Some comments:

Having that said, the weakness of the paper is that it is solely based on an in vitro approach. Engineering an in vivo NPC that would only contain a single engineered FG-Nup is challenging. Although seemingly feasible in yeast if done gradually, it would be lots of work and thus beyond the scope of the present study. However, one really wishes to know if a yeast strain with all the same FG repeats would be viable or alternatively, gradually develop fitness defects once native FG-Nups are changed one by one. You could at least refer to Strawn et al, NCB 2004 in the discussion. I understand that this is not the exact same idea. It is still interesting in this context that many repeats can be removed in vivo and cells are still viable.

We included the Strawn reference as suggested. The genetic experiments would actually address two questions, one being how much FG mass is required? And the other, which mix of FG domain types is optimal for fitness?

Indeed, yeast contains different sets of FG domains, very cohesive and non-charged ones (e.g., from Nup100, Nup116, Nup49, Nup54), but also domains with highly charged and rather non-cohesive spacers (Nsp1, Nup159, Nup2, Nup1). We see a function not only for the cohesive but also for the non-cohesive ones. The latter might pose additional filter layers or fine-tune the tightness of the cohesive layers. Too much cohesive FG mass (as a result of FG domain exchange experiments) might be deleterious and lead to NPCs that are too restrictive for supporting high-capacity nuclear transport. More complex experimental systems are needed to address how the different FG domains cooperate with each other. Considering this, we added the following paragraph to the Perspectives:

Nup98 FG phases function in NPCs that contain a considerable mass of less cohesive or even non-cohesive FG domains. These additional domains are typically localized at the cytoplasmic or nuclear periphery of NPCs and probably represent platforms for rapid disassembly of NTR-cargo complexes. They might also form additional filter zones to fine-tune the selectivity of NPCs. Cohesive FG-FG interactions compete with NTR-binding to FG motifs. Non-cohesive FG domains might therefore be more efficient in NTR-capture, thus functioning as “collectors” for incoming NTR-cargo complexes and increasing the capacity of NPCs for active transport. Indeed, it will be very exciting to reconstitute the cooperation of Nup98 FG domains with other FG domains and test those assumptions. A layer-wise FG phase assembly system might be the first step in this direction.

Style: As pointed out above, this reviewer appreciates the middle part of the manuscript, but it does interrupt the flow of the paper. I do not have a good suggestion where to place it elsewhere but you may consider trimming it down – or integrate it better.

We agree and have tried to better integrate the part.

Page 13: Why did the authors use an RFP fusion of Ran? This should perfectly work with native RanQ69LDeltaC-GTP, correct? It would be important to know if release becomes faster, consistent with the idea that non-tagged Ran can diffuse into the phase – or not.

We used the Ran-fusion to counter the formal argument that cargo-dissociation from the import happens already inside the FG phase and not concomitantly with an exit from the barrier. Using a non-fused Ran, efflux becomes faster by a factor of two (now in Supp.Fig.S8), consistent with the idea of non-fused Ran diffusing more rapidly and being better able to access the FG phase’s interior. It should be noted, though, that fluorophore-labeled RanGTP hardly enters the FG-phase (see Fig.6F) – even though the attached fluorophore itself favors the FG-phase. Free, unlabeled RanGTP should therefore show an even more prominent exclusion from the FG phase.

Introduction on page 4: While the 3rd paragraph emphasizes the remarkable sequence conservation of Nup98FG, the 4th paragraph states that ‘substantial progress has been hampered by the sequence diversity within given Nup98FG domains’. For readability, you may want to include an additional sentence explaining the concept of comparing sequences of FG repeats within the same protein to contrast previous paragraph better (most people think in terms of comparing sequences across proteins).

Agreed. We changed the sentence to read: “*However, substantial progress has been hampered by any given Nup98 FG repeat domain being irregular along its sequence, with variable FG motifs, inter-FG distances, and inter-FG spacers.*”.

Related to that, if the different properties of individual repeats are conserved across different species but a simple repeat recapitulates nuclear transport, what could be the selection pressure?

Excellent question. We would assume that the bulk phase properties impose just one constraint (keeping, e.g., the overall FG density, “best” FG motifs, and amino acid composition constant). Another constraint might be how to fill the central NPC channel evenly; this could explain why the FG density within a given domain increases with distance from their anchor points. Otherwise, we think that additional molecular interactions must account for the remarkable conservation, in particular of the Nup98 FG domain amongst vertebrates. These could relate to

special requirements when transporting very large cargoes, linear motifs targeted by post-translational modifications or being required for NPC assembly.

Some of the authors have previously shown that MacNup98 is bit special in the sense that it contains many glycines and forms a very tight hydrogel already at comparably low intra-particle FG domain concentrations (eLife paper from 2013). Why is it a well-justified choice for this study that hopefully allows for generic conclusions? The statement that it is well-characterized is understandable. However, it makes the reader wonder about the relevance, because the source species is not exactly a main stream model organism. A bit of additional introduction may be helpful.

Well, our choice was pragmatic and also driven by trying to avoid the expected experimental difficulties with fungal and animal Nup98 FG domains. Animal Nup98 FG domains (from mammals, frogs, insects, nematodes) carry O-GlcNAc-modifications, which tune the cohesiveness of the domain. Lack of this modification causes a very restrictive phase, while over-modification impedes phase separation (Labokha et al., 2013). This additional dimension complicates the analysis considerably.

The glycosylation density is adjusted in cells through antagonistic activities (OGT and deglycosylating enzymes, with NTRs probably masking some of the modification sites). This is very hard to recapitulate with the purified enzymes.

Furthermore, any change in sequence might also alter glycosylation sites. For sure, having one sugar per repeat (in a perfectly repeated sequence) would imply over-glycosylation and result in an FG domain of low cohesiveness.

The yeast Nup98-like FG domains from Nup100 and Nup116 are rather NQ-rich and prone to a prion- and amyloid-like behavior. We expect any regularization of their sequences will exaggerate their amyloid propensity. Another complication is that N and Q are encoded by just two codons each, meaning that perfectly repeated versions will be encoded by rather repetitive DNA sequences that are hard to synthesize and maintain as stable plasmids.

The FG domain from the *Tetrahymena thermophila* macronuclear Nup98A shows none of these complications and is by all means very well behaved. We have added a few words (in the first paragraph of Results and Discussion) to explain some of these considerations.

The authors state that they expect Hoechst dyes to become useful for staining FG-phases in vitro, but they also show that even a single dye attached to GFP has a quite massive effect. This does not come across very sound and maybe explained better. What type of experiments could be done despite the fact that the dye massively changes the given phase?

A covalently linked fluorophore indeed increases the partition coefficient of GFP (and probably of other molecules) in an FG phase. The Hoechst dyes are not covalently coupled. They are just added at a low concentration to the mixture. Indeed, we found that they hardly change the FG phase behavior and the partition coefficient of other mobile species (see Figure 4F), as long as they are not irradiated by UV with less than 400nm. We clarified the text accordingly.

This reviewer does not like the term ‘authentic NPC’, because it inversely suggests that phase separated FG-bodies are ‘non-authentic NPCs’. Why can’t it just be an ‘NPC’?

Agreed. We now call them simply ‘NPCs’.

Reviewer #3 (Remarks to the Author):

The underlying mechanisms governing the selective barrier imposed by the nuclear pore complex (NPC) remains to be fully understood. This paper takes an innovative protein-engineering approach to define a “simple” sequence that can recapitulate the fundamental selective sorting properties of the NPC in vitro. By beginning with a Nup98 FG-domain from Tetrahymena, the authors sequentially minimize the sequence to a near perfect repeating GLFG peptide. A critical control is a similar peptide lacking proline residues. Using these reagents, they demonstrate that they can form cohesive networks in vitro that phase-separate into condensates that can recapitulate both active import and export while excluding an inert mCherry protein. Along the way, they also interrogate how these condensates interact with a series of fluorescent probes. The major advance here is that they demonstrate that DAPI can crosslink the FG-network, which could provide a useful tool for the field. In general,

the data are of high quality and are rigorous. The manuscript is written well although it is a bit disjointed due to the “detour” exploring interactions with probes. It should make a valuable contribution to Nature Communications as it will provide valuable concepts and tools for multiple fields.

Thank you very much!

Major Point 1: The paper is written with considerable tunnel vision as to how the NPC acts as a selective barrier as it is solely focused on the importance of cohesive interactions. It would be nice if the authors at least considered that there are non-cohesive FG-networks in the NPC that are known to be important functionally. No additional data is required, but a discussion of the disconnect, for example, between the ability of the GLFG-domain to exclude mCherry in vitro and the observation that mCherry can freely pass through the NPC in vivo would seem warranted, particularly in the context of work defining the NPC barrier in vivo as being “soft” (e.g. Timney, JCB).

We added a section to discuss non-cohesive interactions as suggested. We wish to add, though, that mCherry passes NPCs 700 times more slowly than NTF2 (Frey et al., 2018; see also Figure 5). We wouldn't call this a free passage. In fact, the retention of 25-30 kDa-sized proteins is functionally important because it allows to build up a steep nucleocytoplasmic RanGTP-gradient. We have re-written the first part of the introduction to clarify these points.

Major Point 2: Hydrogel versus liquid? A description of the condensates in terms of their physical characteristics would be helpful.

We already estimated diffusion coefficients of the FG domains within the phases and have expanded on this.

Although it is acknowledged that there is a continuum of physical states between liquids and gels, there is also the concept that cohesive interactions can transition or “age” between liquids and more solid states, the latter being less functional/pathological. The photobleaching in Fig. 6D may, for example, be indicating that the GLEBS domain drives a gel-like state. Again, I don't think any additional experimentation is needed, but some discussion of this in the context of the phase-separation literature should be considered.

We have observed that the Nup116 FG phase accumulates Thioflavin-T positive structures over the course of ≥ 10 hours (Supp.Fig. S1), which can be interpreted as aging. Apart from this, we haven't seen any aging effects in terms of FG phase selectivity. Given that NPCs are long-lived structures, there was probably selective pressure against age-related deterioration of barrier function. We do fully agree that this is a very important topic, in particular for defining the position of FG phases within the universe of biological condensates. However, we feel that we cannot adequately cover this topic within a few sentences and therefore wish to postpone the discussion for a comprehensive review (also given that the present manuscript is already very long).

Reviewers' Comments:

Reviewer #1:

Remarks to the Author:

The revised manuscript has improved in terms of presentation and clarity, but my major concerns remain about the novelty of the method and the conclusions being drawn.

As stated in my previous comments, both in vitro mimic of the NPC's selectivity barrier by using FG domains to form hydrogels and then tests of importins and exportins interacting with this hydrogel have been fully studied in their previous papers (Science , 2006, 314:815; Cell, 2007, 130:512). Since the author denied it, this reviewer has to include figure 4 of that Science paper and also the abstract of 2007 Cell paper in the attachment, in which clearly the interactions between NTRs (importins & exportins) and hydrogel have been fully studied and the conclusion that FG-hydrogel can mimic the NPC's permeability barrier has already been drawn. Thus, compared to these studies published more than ten years ago, small increments presented in the current manuscript would unlikely bring anything new or impactful into the field of NPC and nucleocytoplasmic transport.

Reviewer #2:

Remarks to the Author:

The authors have carefully addressed my comments. I strongly recommend this manuscript for publication in Nat Comm.

Reviewer #3:

Remarks to the Author:

I was supportive of this paper after the first submission. I appreciate that the authors took the time to address my few comments.